# Decomposed Mutual Information Optimization for Generalized Context in Meta-Reinforcement Learning

**Yao Mu**
The University of Hong Kong
muyao@connect.hku.hk

**Yuzheng Zhuang**
Huawei Noah's Ark Lab
zhuangyuzheng@huawei.com

**Fei Ni**
Tianjin University
fei_ni@tju.edu.cn

**Bin Wang**
Huawei Noah's Ark Lab
wangbin158@huawei.com

**Jianyu Chen**
Tsinghua University
jianyuchen@tsinghua.edu.cn

**Jianye Hao**
Huawei Noah's Ark Lab
haojianye@huawei.com

**Ping Luo** [*]
The University of Hong Kong
pluo@cs.hku.hk

## Abstract

Adapting to the changes in transition dynamics is essential in robotic applications. By learning a conditional policy with a compact context, context-aware meta-reinforcement learning provides a flexible way to adjust behavior according to dynamics changes. However, in real-world applications, the agent may encounter complex dynamics changes. Multiple confounders can influence the transition dynamics, making it challenging to infer accurate context for decision-making. This paper addresses such a challenge by **DecO**mposed **M**utual **IN**formation **O**ptimization (DOMINO) for context learning, which explicitly learns a disentangled context to maximize the mutual information between the context and historical trajectories, while minimizing the state transition prediction error. Our theoretical analysis shows that DOMINO can overcome the underestimation of the mutual information caused by multi-confounded challenges via learning disentangled context and reduce the demand for the number of samples collected in various environments. Extensive experiments show that the context learned by DOMINO benefits both model-based and model-free reinforcement learning algorithms for dynamics generalization in terms of sample efficiency and performance in unseen environments. Open-sourced code is released on our homepage.

## 1 Introduction

Dynamics generalization in deep reinforcement learning (RL) investigates the problem of training a RL agent in a few kinds of environments and adapting across unseen system dynamics or structures, such as different physical parameters or robot mythologies. Meta-Reinforcement Learning (Meta-RL) has been proposed to tackle the problem by training on a range of tasks, and fast adapting to a new task with the learned prior knowledge. However, training in meta-RL requires orders of magnitudes more samples than single-task RL since the agent not only has to learn to infer the change of environment but also has to learn the corresponding policies. Context-aware meta-RL methods take a step further and show promising potential to capture local dynamics explicitly by learning an

---

[*]Ping Luo is the corresponding author. Yao Mu and Fei Ni conducted this work during the internship in Huawei Noah's Ark Lab.

36th Conference on Neural Information Processing Systems (NeurIPS 2022).

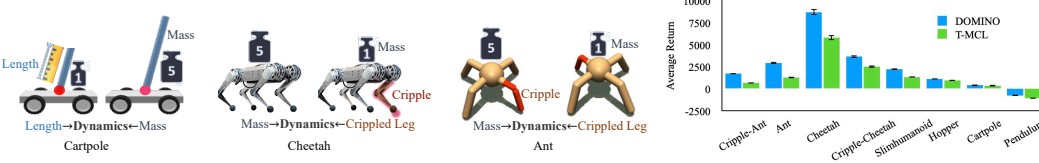



(a) Example of the multi-confounded environments      (b) Performance comparison



Figure 1: Generalization with complex dynamics changes. The transition dynamics of the robot may simultaneously influenced by multiple confounders, such as mass (⬛), length of leg (▬), or a crippled leg (🦵). In real-world situations, all the possible confounders may change simultaneously, which bring challenges to robotic dynamics generalization. DOMINO addresses such problem by decomposed MI optimization and achieves the state-of-the-art performance.

additional context vector from historical trajectories [1–4]. The historical trajectories are sampled from the joint distribution of multiple confounders, which are the key factors that cause the dynamics changes. Accordingly, if multiple confounders affect the dynamics simultaneously, the state transition distribution will become highly multi-modal, leading to challenges in extracting accurate context.

Recent advanced context-aware meta-RL methods [5–8] further improve meta-RL via contrastive learning, which optimizes the InfoNCE bound [9] of the mutual information in essence. These methods show a promising improvement in entangled context learning, which performs well in single confounded environments. However, as demonstrated in Figure 1a, in real-world situations for robotic applications with partially unspecified dynamics, the transition dynamics can be influenced by multiple confounders simultaneously, such as mass changes, damping, friction, or malfunctional modules like a crippled leg. For example, when a transportation robot is working in the wild, the load will dynamically change as the task progresses, while the humidity and roughness of the road also may vary. Moreover, some works also construct a confounder set for unsupervised RL environment generalization[10–15]. RIA [16] also constructs confounder sets with multiple confounders for unsupervised dynamics generalization. Such changeable environments bring great challenges to the robot for capturing contextual information, which motivates our study.

**Contribution.** In this paper, we give a theoretical analysis which demonstrates that when the number of confounders increases, InfoNCE will be a loose bound of mutual information (MI) with the samples in limited seen environments, which is called MI underestimation [17]. To tackle this problem, we propose a **D**ec**O**mposed **M**utual **IN**formation **O**ptimization (DOMINO) framework for context learning in meta-RL. The context encoder aims to embed the past state-action pairs into disentangled context vectors and is optimized by maximizing the mutual information between the disentangled context vectors and historical trajectories while minimizing the state transition prediction error. DOMINO decomposes the full MI optimization problem into a summation of $N$ smaller MI optimization problems by learning disentangled context. We then theoretically prove that DOMINO could alleviate the underestimation bias of the InfoNCE and reduce the demand for the samples collected in various environments [18, 19]. Last, with the learned disentangled context, we further develop the context-aware model-based and model-free algorithms to learn the context-conditioned policy and illustrate that DOMINO can consistently improve generalization performance in both ways to overcome the challenge of multi-confounded dynamics.

Extensive experiments demonstrate that DOMINO benefits meta-RL on both the generalization performance in unseen environments and sample efficiency during the training process under the challenging multi-confounded setting. For example, as show in Figure 1b, it achieves 1.5 times performance improvement to T-MCL [3] in the Cheetah domain and 2.6 times performance improvement to T-MCL in the Crippled-Ant domain. Visualization of the learned context demonstrates that the disentangled context generated by DOMINO under different environments could be more clearly distinguished in the embedding space, which indicates its advantage to extract high-quality contextual information from the environment.

## 2 Related Work

### 2.1 Meta-Reinforcement Learning

Meta-RL extends the framework of meta-learning [20, 21] to reinforcement learning, aiming to learn an adaptive policy being able to generalize to unseen tasks. Specifically, meta-RL methods

learn the policy based on the prior knowledge discovered from various training environments and reuse the policy to fast adapt to unseen testing environments after zero or few shots. Gradient-based meta-RL algorithms [22–25] learn a model initialization and adapt the parameters with few policy gradient updates in new dynamics. Context-based meta-RL algorithms [1–4] learn contextual information to capture local dynamics explicitly and show great potential to tackle generalization tasks in complicated environments. Many model-free context-based methods are proposed to learn a policy conditioned on the latent context that can adapt with off-policy data by leveraging context information and is trained by maximizing the expected return. PEARL [1] adapts to a new environment by inferring latent context variables from a small number of trajectories. Recent advanced methods further improve the quality of contextual representation leveraging contrastive learning [5–8]. Unlike the model-free methods mentioned above, context-aware world models are proposed to learn the dynamics with confounders directly. CaDM [26] learns a global model that generalizes across tasks by training a latent context to capture the local dynamics. T-MCL [4] combines multiple-choice learning with context-aware world model and achieves state-of-the-art results on the dynamics generalization tasks. RIA [16] further expands this method into unsupervised setting without environment label by intervention, and enhances the context learning via MI optimization.

However, existing context-based approaches focus on learning entangled context, in which each trajectory is encoded into only one context vector. In a multi-confounding environment, learning entangled contexts requires orders of magnitude higher samples to capture accurate dynamics information. To tackle this challenge, different from RIA [16] and T-MCL [4] , DOMINO infers several disentangled context vectors from a single trajectory and divides the whole MI optimization into the summation of smaller ones. The proposed decomposed MI optimization reduces the amount of demand for diverse samples and thus improves the generalization of the policy to overcome the adaptation problem in multi-confounded unseen environments.

## 2.2 Mutual Information Optimization for Representation Learning

Representation learning based on mutual information (MI) maximization has been applied in various tasks such as computer vision [27, 28], natural language processing [29, 19], and RL [30], exploiting noise-contrastive estimation (NCE) [31], InfoNCE [9] and variational objectives [32]. InfoNCE has gained recent interest with respect to variational approaches due to its lower variance [33] and superior performance in downstream tasks. However, InfoNCE may underestimate the true MI, given that it is limited by the number of samples. To tackle this problem, DEMI [17] first scaffolds the total MI estimation into a sequence of smaller estimation problems. In this paper, since the confounders in the real world are commonly independent, we simplify the complexity of mutual information decomposition and eliminate the need to learn conditional mutual information as a sub-term, assuming that multiple confounders are independent of each other.

# 3   Preliminaries

We consider standard RL framework where an agent optimizes a specified reward function through interacting with an environment. Formally, we formulate our problem as a Markov decision process (MDP) [34], which is defined as a tuple $(\mathcal{S}, \mathcal{A}, p, r, \gamma, \rho_0)$. Here, $\mathcal{S}$ is the state space, $\mathcal{A}$ is the action space, $p(s'|s, a)$ is the transition dynamics, $r(s, a)$ is the reward function, $\rho_0$ is the initial state distribution, and $\gamma \in [0, 1)$ is the discount factor. In order to address the problem of generalization, we further consider the distribution of MDPs, where the transition dynamics $p_{\tilde{u}}(s'|s, a)$ varies according to multiple confounders $\tilde{u} = \{u_0, u_1, \ldots, u_N\}$. The confounders can be continuous random variables, like the mass, damping, random disturbance force, or discrete random variables, such as one of the robot's leg is crippled. We assume that the true transition dynamics model is unknown, but the state transition data can be sampled by taking actions in the environment. Given a set of training setting sampled from $p(\tilde{u}_{\text{train}})$, the meta-training process learns a policy $\pi(s, c)$ that adapts to the task at hand by conditioning on the embedding of the history of past transitions, which we refer as context $c$. At test-time, the policy should adapt to the new MDP under the test setting $\tilde{u}_{\text{test}}$ drawn from $p(\tilde{u}_{\text{test}})$.

Our goal is to learn a policy to maximizing the expected return $\mathcal{R}_{\text{train}}$ condition on the context $c$ which is encoded from the sequences of current state action pairs $\{s_\tau, a_\tau, s_{\tau+1}\}_{\tau=t-H}^{t}$ in several training

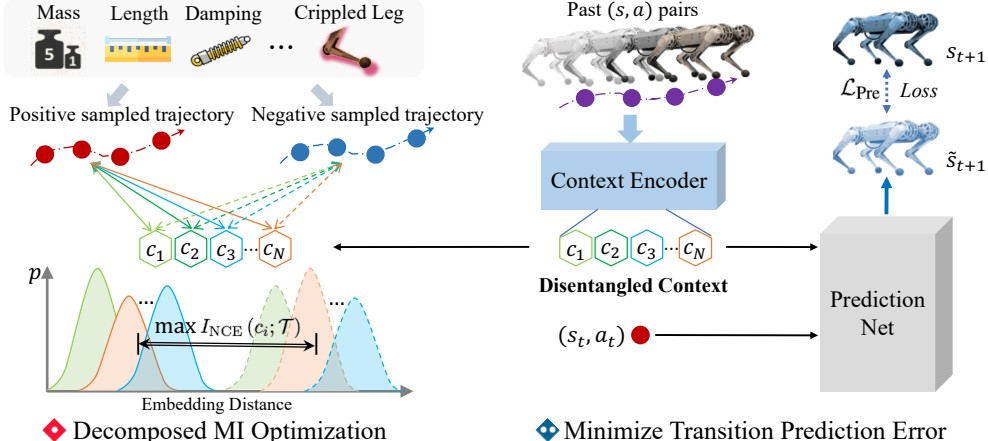

Figure 2: The overall framework of DOMINO (⬥). The context encoder embeds the past state-action pairs (•••) into disentangled context vectors. The disentangled context vectors (◯) are learned via the decomposed mutual information optimization (◆) while minimizing the state transition prediction error (⬥). **Minimize transition prediction error**: With the current state-action pair and the learned context vectors, future state can be predicted by the prediction network. The gradient of the prediction error will be used to update both the context encoder and the prediction network. **Decomposed MI Optimization**: we optimize the MI between the learned context and the historical trajectories under the same confounder setting via maximizing the InfoNCE bound, which aims to minimize the embedding distance between the positive sampled trajectories (•••) and the context, while maximizing the embedding distance between the negative sampled trajectories (•••) and the context.

scenarios and enable it to perform well and achieve a high expected return $\mathcal{R}_{test}$ in test scenarios never seen before.

$$\max_{\pi} \left\{ \mathcal{R}_{\#} = \mathbb{E}_{\tilde{u} \sim p(\tilde{u}_{\#})} \left[ \sum_{t=0}^{\infty} \gamma^t r\left(\mathbf{s}_t, \mathbf{a}_t\right) \right] \right\}, \quad a_t \sim \pi(s_t, c), \quad \# = \{\text{"train" or "test"}\} \quad (1)$$

## 4 Decomposed Mutual Information Optimization for Context Learning

In this section, we first provide a theoretical analysis to show why the multi-confounded environments are more challenging. We find that when the number of confounders increases, the InfoNCE will be a loose bound of MI with the samples in limited seen environments, resulting in the underestimation of MI. To solve such a problem, we develop the DOMINO framework to learn disentangled context by decomposed MI optimization. We theoretically illustrate that the decomposed MI optimization can alleviate the underestimation of MI and reduce the demand for the number of samples. The disentangled context $c = \{c_0, c_1, \ldots, c_N\}$ is embedded by the context encoder with parameter $\varphi$ from the past state-action pairs $\tau^* = \{s_l^*, a_l^*\}_{l=t-H}^{t-1}$ in current episode. DOMINO explicitly maximizes the MI between the context $c$ and the historical trajectories $\mathcal{T} = \{\tau^i\}_{i=1}^{M}$ ($\tau^i = \{s_l^i, a_l^i\}_{l=0}^{T}$) collected based on the combination of multiple confounders $\tilde{u} = \{u_0, u_1, \ldots, u_N\}$ as same as the current confounder setting, while minimizing the state transition prediction error conditioned on the learned context. We solve the MI optimization problem by optimizing the InfoNCE lower bound on MI [9], which can be viewed as a contrastive method for the MI optimization, and decompose the full MI optimization into smaller ones to alleviate the underestimation of the mutual information and reduce the demand for the number of samples collected in various environments.

### 4.1 InfoNCE Bound for Mutual Information Optimization

InfoNCE bound $I_{\text{NCE}}(x; y)$ is a lower bound of the mutual information $I(x; y)$, where NCE stands for Noise-Contrastive Estimation, is a type of contrastive loss function used for self-supervised learning. InfoNCE is obtained by comparing pairs sampled from the joint distribution $x, y_1 \sim p(x, y)$ ($y_1$ is

called the positive example) to pairs $x, y_i$ built using a set of negative examples, $y_{2:K} \sim p(y_{2:K}) = \prod_{k=2}^{K} p(y_k)$:

$$I(x;y) \geq I_{\text{NCE}}(x;y \mid \psi, K) = E \left[ \log \frac{e^{\psi(x,y_1)}}{\frac{1}{K} \sum_{k=1}^{K} e^{\psi(x,y_k)}} \right] \tag{2}$$

where $\psi$ is a function assigning a similarity score to $x, y$ pairs and $K$ denotes the number of samples. Through discriminating naturally the paired positive instances from the randomly paired negative instances, it is proved to bring universal performance gains in various domains, such as computer vision and natural language processing.

**Lemma 1** $I_{\text{NCE}}(X;Y \mid K) \leq I(x;y) \leq \log K$ *is a necessary condition for $I_{\text{NCE}}(X;Y \mid K)$ to be a tight bound of $I(x;y)$. (see proof in Appendix A)*

Some previous context-aware methods learn an entangled context $c$ by maximizing the mutual information between the context $c$ embedded from the past state-action pairs in the current episode, and the historical trajectories $\mathcal{T}$ collected under the same confounder setting as the current episode. They solve this problem by maximizing the InfoNCE lower bound on $I_{\text{NCE}}(c; \mathcal{T})$, which can be viewed as a contrastive estimation [9] of $I_{\text{NCE}}(c; \mathcal{T})$, and obtain promising improvement in single-confounded environment. However, according to Lemma 1, the $I_{\text{NCE}}(c; \mathcal{T})$ may be loose if the true mutual information $I(c; \mathcal{T})$ is larger than $\log K$, which is called underestimation of the mutual information. Therefore, to make the InfoNCE bound to be a tight bound of $I(c; \mathcal{T})$, the minimum number of samples is $e^{I(c;\mathcal{T})}$. In real-world robotic control tasks, the dynamics of the robot is commonly influenced by multiple confounders $\tilde{u} = \{u_0, \ldots, u_i, \ldots, u_j, \ldots, u_N\}$ simultaneously, under the assumption that the confounders are independent (such as mass and damping), the mutual information between the historical trajectories $\mathcal{T}$ and the context $c$ can be derived as

$$\begin{aligned} I(c;\mathcal{T}) &= \mathbb{E}_{p(\tau,c)} \left\{ \log \frac{p(\mathcal{T} \mid c)}{p(\mathcal{T})} \right\} = \mathbb{E}_{p(\tau,c)} \log \left\{ \frac{\int p(\mathcal{T} \mid \tilde{u}) p(\tilde{u} \mid c) \mathrm{d}\tilde{u}}{p(\mathcal{T})} \right\} \\ &\geq \mathbb{E}_{p(\tau,c)p(\tilde{u}|c)} \left\{ \log \frac{p(\mathcal{T} \mid \tilde{u})}{p(\mathcal{T})} \right\} = I(\tilde{u};\mathcal{T}) \overset{u_i \perp u_j}{\Longrightarrow} \sum_{i=0}^{N} I(u_i;\mathcal{T}) \end{aligned} \tag{3}$$

As the number of confounders increases, the lower bound of $I(c; \mathcal{T})$ will become larger, and the necessary condition for $I_{\text{NCE}}(\mathcal{T}; c \mid K)$ to be a tight bound of $I(c; \mathcal{T})$ will become more difficult to satisfy. Since $I(c; \mathcal{T}) \geq \sum_{i=0}^{N} I(u_i; \mathcal{T})$, to let the necessary consition satisfied, the amount of data $K$ must be larger than $e^{\sum_{i=0}^{N} I(u_i;\mathcal{T})}$ according to Lemma 1. Thus the demand for data increases significantly. Since the confounders are commonly independent in real-world, can we relax this condition by learning disentangled context vectors instead of entangled context intuitively?

## 4.2 Decomposed MI Optimization

If the context vectors $c = \{c_0, c_1, \ldots, c_N\}$ can be independent, then we can ease this problem by applying the chain rule on MI to decompose the total MI into a sum of smaller MI terms, i.e.,

$$I_{\text{NCE}}(c; \mathcal{T} \mid K) = \sum_{i=0}^{N} \{I_{\text{NCE}}(c_i; \mathcal{T} \mid K)\} \leq N \log K \tag{4}$$

**Theorem 1** *If the context vectors $\{c_0, c_1, \ldots, c_N\}$ can be independent, then the necessary condition for $I_{\text{NCE}}(c; \mathcal{T})$ to be a tight bound can be relaxed to $I(c; \mathcal{T}) \leq N \log K = \log K^N$. Thus, the need of the number of samples can be reduced from $K \geq e^{I(c;\mathcal{T})}$ to $K \geq e^{\frac{1}{N} I(c;\mathcal{T})}$.*

Inspired by Theorem 1, we intuitively learn disentangled context vectors and maximize the mutual information between the historical trajectories $\mathcal{T}$ and the context vectors $\{c_0, \ldots, c_N\}$ while minimizing the $I_{\text{NCE}}$ between the context vectors, i.e., to maximize the $\mathcal{L}_{\text{NCE}}$

$$\mathcal{L}_{\text{NCE}}(\varphi, w) = \sum_{i=0}^{N} I_{\text{NCE}}(c_i; \mathcal{T}) - \sum_{j=0}^{N} \sum_{i=0, i \neq j}^{N} I_{\text{NCE}}(c_i; c_j) \tag{5}$$

where the $I_{\text{NCE}}(c_i; \mathcal{T})$ can be obtained with the positive trajectory $\tau^+$ and negative trajectories $\{\tau_k^-\}_{k=2}^{K}$, i.e.,

$$I_{\text{NCE}}(c_i; \mathcal{T}) = E\left[\log \frac{e^{\psi(c_i, h_w(\tau^+))}}{\frac{1}{K}\left(\sum_{k=2}^K e^{\psi(c_i, h_w(\tau_k^-))} + e^{\psi(c_i, h_w(\tau^+))}\right)}\right] \tag{6}$$

The context $c = \{c_0, \ldots, c_i, \ldots, c_N\}$ $(c_i \in \mathbb{R}^m)$ is encoded from the past state action pairs $\tau^*$ in current episode by $g_\varphi(\cdot)$. Both the positive trajectory (collected in same setting of the confounders) and negative trajectory (collected in different setting of the confounders) are encoded to $\mathbb{R}^m$ by $h_w(\cdot)$. The critic function $\psi(\cdot, \cdot)$ measures the cosine similarity between inputs by dot product after normalization. Under the assumption that the setting of confounders will not change in one episode, to obtain the $I_{\text{NCE}}(c_i; c_j)$, we use $c_j^+$ sampled from same episode like $c_i$ as the positive example, and use $c_j^-$ sampled from different episode as the negative example. Thus the $I_{\text{NCE}}(c_i; c_j)$ can be derived as

$$I_{\text{NCE}}(c_i; c_j) = E\left[\log \frac{e^{\psi(c_i, c_j^+)}}{\frac{1}{K}\left(\sum_{k=2}^K e^{\psi(c_i, c_j^{k-})} + e^{\psi(c_i, c_j^+)}\right)}\right] \tag{7}$$

Then, the future state $s_{t+1}$ can be predicted with the the current state $s_t$, action $a_t$ and the disentangled context vectors $\{c_{0_t}, \ldots, c_{N_t}\}$ by the state transition prediction network $f_\phi(\cdot)$. We aim to minimize the prediction loss, which is equal to maximizing

$$\mathcal{L}_{\text{Pre}}(\varphi, \phi) = E_{\tau^* \sim \mathcal{B}}\left[-\frac{1}{H}\sum_{\lambda=t}^{t+H-1} \log f_\phi\left(s_{\lambda+1} \mid s_\lambda, a_\lambda, (c_{\lambda_0}, \ldots, c_{\lambda_N})\right)\right], \tau^* = \{s_l^*, a_l^*\}_{l=t-H}^{t-1} \tag{8}$$

where $\mathcal{B}$ is the training set, and $H$ is the prediction horizon. The whole framework of DOMINO is demonstrated in Figure 2 and the overall objective function of DOMINO is

$$\mathcal{L}(\varphi, w, \phi) = \mathcal{L}_{\text{Pre}}(\varphi, \phi) + \mathcal{L}_{\text{NCE}}(\varphi, w) \tag{9}$$

### 4.3 Combine DOMINO with Downstream RL Methods

**Combination with Model-based RL.** With DOMINO we can learn the context encoder and the context-aware world model together. First, the past state-action pairs are encoded into the disentangled context vectors by the context encoder. According to the learned context, the transition prediction network predicts the future states of different actions. In particular, we use the cross entropy method (CEM) [35], a typical neural model predictive control (MPC) [36] method, to select actions, in which several candidate action sequences are iteratively sampled from a candidate distribution, which is adjusted based on best-performing action samples. The optimal action sequence $\boldsymbol{a}_{t:t+T} \doteq \{\boldsymbol{a}_t, \ldots, \boldsymbol{a}_{t+T}\}$ can be obtained by

$$\text{argmax}_{\boldsymbol{a}_{t:t+T}} \sum_{\lambda=t}^{t+T} \mathbb{E}_{\tilde{f}}\left[r\left(\boldsymbol{s}_\lambda, \boldsymbol{a}_\lambda\right)\right], \quad \tilde{f} = \Pr\left(s_{t+1} \mid s_t, a_t, c_{t_0}, \ldots, c_{t_N}\right) \tag{10}$$

Then, we use the mean value of adjusted candidate distribution as action and re-plan at every timestep. We provide detailed algorithm pseudo-code in the Appendix B.1. As for the adaptation process, the policy and context encoder zero-shot adapts to the unseen confounders setting $u_{test}$, and we use the same adaptive planning method as T-CML[3], which selects the most accurate prediction head over a recent experience condition on the inferred context. The details is introduced in Appendix D.4

**Combination with Model-free RL.** Previous works show that a policy learned by model-free method can be more robust to dynamics changes when it takes the contextual information as an additional input [37–39]. Motivated by this, we investigate whether the context encoder learned by DOMINO can be used as a plug-and-play module to improve the final generalization performance of model-free RL methods. We concatenate the disentangled context encoded by a pre-trained context encoder from DOMINO and the current state-action pairs, and learn a conditional policy $\pi\left(a_t | s_t, c_0, \ldots, c_N\right)$. We use the Proximal Policy Optimization (PPO) method to train the agent [40], which learns the policy by maximizing

$$\hat{\mathbb{E}}_t\left[\frac{\pi\left(a_t \mid s_t, c_{t_0}, \ldots, c_{t_N}\right)}{\pi_{\theta_{\text{old}}}\left(a_t \mid s_t, c_{t_0}, \ldots, c_{t_N}\right)}\hat{A}_t - \beta\text{KL}\left[\pi_{\theta_{\text{old}}}\left(\cdot \mid s_t, c_{t_0}, \ldots, c_{t_N}\right), \pi\left(\cdot \mid s_t, c_{t_0}, \ldots, c_{t_N}\right)\right]\right] \tag{11}$$

where $\hat{A}_t$ is the estimation of the advantage function at timestep $t$. We provide detailed pseudo-code in the Appendix B.2.

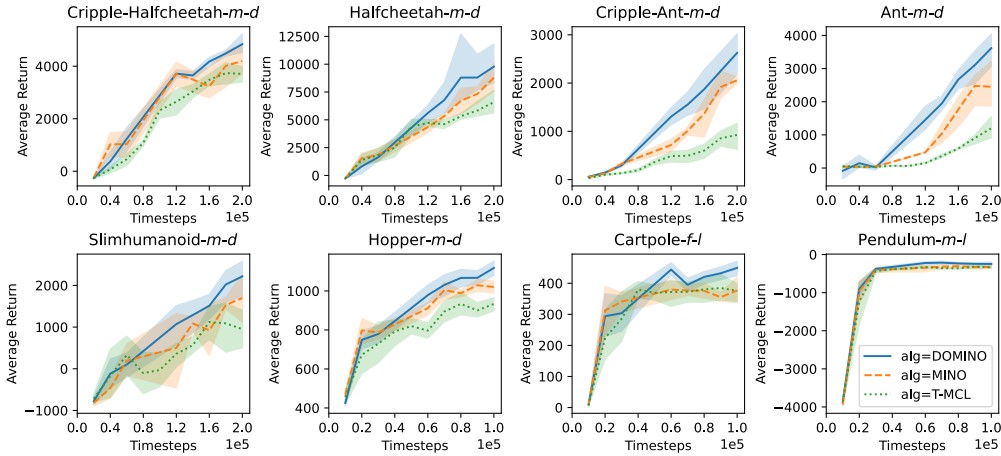

Figure 3: The average returns of the model-based methods in training environments (over 8 seeds).

## 5 Experiments

In this section, we evaluate the performance of our DOMINO method to answer the following questions: **(1)** Can DOMINO help the model-based RL methods overcome the multi-confounded challenges in dynamics generalization (see comparison with ablation in Figure 3 and Figure 5)? **(2)** Can the context encoder learned by DOMINO be used as a plug-and-play module to improve the generalization abilities of model-free RL methods in multi-confounded environments (see comparison with ablation in Table 1 and Table 2)? **(3)** Can the proposed decomposed MI optimization benefit the forward prediction of the world model? (see Figure 6) **(4)** Does the disentangled context extract more meaningful contextual information than entangled context (see Figure 7)?

### 5.1 Setups

We demonstrate the effectiveness of our proposed method on 8 benchmarks, which contain 6 typical robotic control tasks based on the MuJoCo physics engine [41] and 2 classical control tasks (CartPole and Pendulum) from OpenAI Gym [42]. Different from previous works, all the environments are influenced by multiple confounders simultaneously. In our experiments, we modify multiple environment parameters at the same time (e.g., mass, length, damping, push force, and crippled leg) that characterize the transition dynamics. The robotic control tasks contain 4 environments (Hopper, HalfCheetah, Ant, SlimHumanoid) affected by multiple continuous confounders and 2 more difficult environments (Crippled Ant and Crippled HalfCheetah) affected by both continuous and discrete confounders. The detailed settings are illustrated in Appendix C (Table 3). We implement these environments based on the publicly available code provide by [43, 4], and we also open-source the code of the multiple-confounded environments[2]. For both training and testing phase, we sample the confounders at the beginning of each episode. During training, we randomly select a combination of confounders from a training set. At test time, we evaluate each algorithm in unseen environments with confounders outside the training range.

### 5.2 Comparison with Model-based Methods

**Baselines.** We consider T-MCL [3] and RIA[16] as the key baselines in comparison with model-based methods, which achieve the state-of-the-art results in zero-shot dynamics generalization tasks. Since RIA doesn't has a adaptive planning process, we provide the DOMINO and T-MCL without adaptive planning to fair compare to the RIA. We also consider an *ablation version* of DOMINO as a baseline (denoted as MINO) to show the effectiveness of the decomposed MI optimization, which optimizes the MI and predicts the future states with an entangled context without decomposition.

Figure 5 shows the generalization performance tested in the unseen environments. The results show that DOMINO surpasses T-MCL in terms of the generalization performance and the learning sample

---

[2]https://anonymous.4open.science/r/Multiple-confounded-Mujoco-Envs-01F3

**Results.** As shown in Figure 9, DOMINO achieves better generalization performance than RIA and TMCL even without the adaptive planning, especially in complex environments like Halfcheetah-$m$-$d$ and Slim-humanoid-$m$-$d$. Figure 3 shows the average return during the learning process in the training environments. The results illustrate that DOMINO learns the policy more efficiently than T-MCL and MINO.

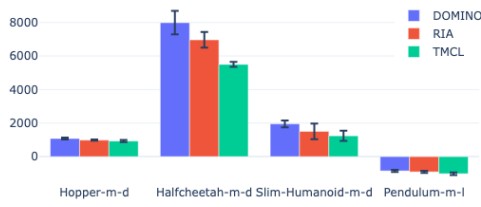

Figure 4: Comparison w/o adaptive planning

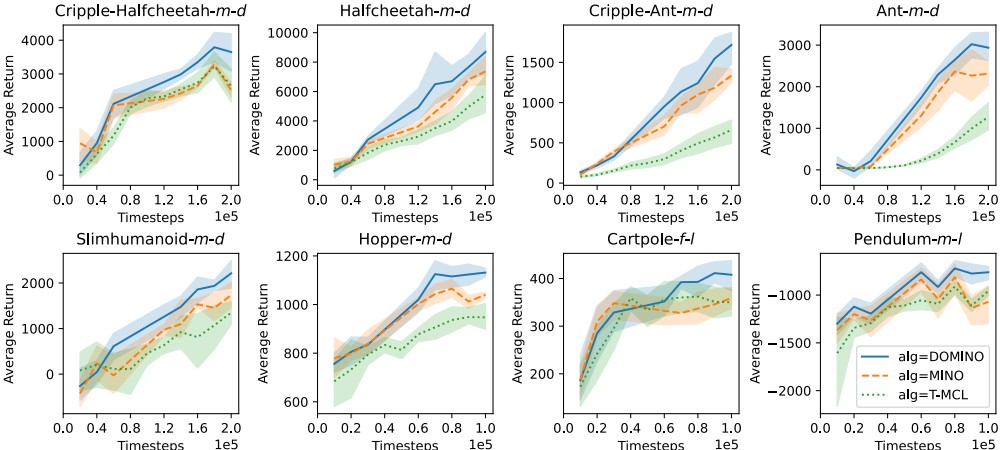

Figure 5: Comparison with *model-based methods* of the generalization performance (average return) in unseen multi-confounded environments (over 8 seeds).

efficiency. This demonstrates that the disentangled context improves the context-aware world model. Especially, the performance gain becomes much more significant in more complex environments (e.g., long-horizon and high-dimensional domains like Cripple-Ant, Ant, and Hopper). For example, DOMINO achieves about 2.6 times improvement to T-MCL in Cripple-Ant-$m$-$d$, which is one of the most difficult environment, whose leg will randomly be crippled, and its mass and damping will be changed in testing. More details are shown in Appendix D.

### 5.3 Comparison with Model-free Methods

We also verify whether the learned disentangled context is useful for improving the generalization performance of model-free RL methods. Similar to [3, 44], we use the Proximal Policy Optimization (PPO [40]) method to train the agents.

**Baselines**. Our proposed method, which takes the context learned by DOMINO as conditional input (PPO+DOMINO), is compared with several context-conditional policies [45, 39]. Specifically, we consider combining the PPO with the context learned by T-MCL (PPO+T-MCL), which learns the context encoder via a context-aware world model and achieves the state-of-the-art performance on dynamics generalization. We also consider PEARL [45], which learns probabilistic context variable by maximizing the expected returns. We further develop an *ablation version* of DOMINO, which optimize the MI with entangled context (PPO+MINO) as a baseline to illustrate the effectiveness of the decomposed MI optimization. We provide more detailed explanations in Appendix D.

**Results**. Table 1 and Table 2 show the performance of various model-free RL methods on both training and test environments. PPO+DOMINO shows superior performance and shows better generalization performances than previous conditional policy methods, implying that the proposed DOMINO method can extract contextual information more effectively than both the context learned by the model-based method (PPO+T-MCL) and the context learned by the model-free method (PEARL). Furthermore, PPO+DOMINO shows an obvious advantage over PPO+MINO, especially in complex environments, such as HalfCheetah, Ant, and Hopper, which implies that the decomposed

MI optimization improves the context learning significantly. Additionally, the results also show that compared to PEARL, the context learned by T-MCL and DOMINO has better performance which implies that the state transition perdition can help to extract contextual information more effectively.

Table 1: Comparison with *model-free* methods in the multi-confounded environments (over 5 seeds). The transition dynamics will change in both training and test environments in every episode.

| | Cartpole-$f$-$l$ | | Pendulum-$m$-$l$ | | Ant-$m$-$d$ | |
|---|---|---|---|---|---|---|
| | Train | Test | Train | Test | Train | Test |
| PEARL | 197±12 | 175±37 | -1265±173 | -1293±134 | 153±63 | 73±25 |
| PPO+T-MCL | 220±27 | 182±25 | -558±184 | -579±128 | 176±82 | 173±38 |
| PPO+MINO | 267±18 | 234±46 | -497±162 | -526±219 | 194±95 | 184 ± 47 |
| PPO+DOMINO | **299**±23 | **283**±68 | **-405**±139 | **-436**±146 | **227**±86 | **216**±52 |
| | Halfcheetah-$m$-$d$ | | Slimhumanoid-$m$-$d$ | | Hopper-$m$-$d$ | |
| | Train | Test | Train | Test | Train | Test |
| PEARL | 1802±773 | 530±270 | 6947±3541 | 3697±2674 | 934±242 | 874±366 |
| PPO+T-MCL | 2032±688 | 674±395 | 6157±1435 | 4136±1528 | 937±252 | 896±238 |
| PPO+MINO | 1973±563 | 824±498 | 6179±1123 | 4275±1134 | 1109±349 | 964±323 |
| PPO+DOMINO | **2472**±803 | **1034**±476 | **7825**±1256 | **5258**±1039 | **1409**±254 | **1137**±335 |

## 5.4 Disentangled Context Analysis

**Prediction errors**. To show that our method indeed helps with a transition prediction, we compare baseline methods with DOMINO in terms of prediction error across 8 environments with varying multiple confounders. As shown in Figure 6, our model demonstrates superior prediction performance, which indicates that the learned context capture better contextual information compare with entangled context (see more results in Appendix E.1).

**Visualization**. We visualize concatenation of the disentangled context vectors learned by DOMINO via t-SNE [46] and compare it with the entangled context learned by T-MCL. As shown in Figure 7, we find that the disentangled context vectors encoded from trajectories collected under different confounder settings could be more clearly distinguished in the embedding space than the entangled context learned by T-MCL. This indicates that DOMINO extracts high-quality task-specific information from the environment compared with T-MCL. We provide more visualization results based on both t-SNE [46] and PCA [47] in Appendix F.2.

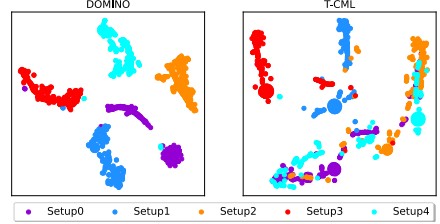
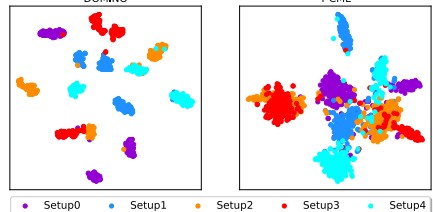
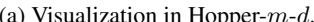

(a) Visualization in Hopper-$m$-$d$.        (b) Visualization in Cripple-Ant-$m$.

Figure 7: t-SNE visualization of context vectors extracted from trajectories collected in various environments. Embedded points from environments with the same confounders have the same color. Hopper setups: *Setup0* ($m = 0.75, d = 1.5$), *Setup1* ($m = 1.5, d = 1.25$), *Setup2* ($m = 0.5, d = 1.25$), *Setup3* ($m = 1.25, d = 0.75$), *Setup4* ($m = 1.0, d = 1.5$); Cripple-Ant setups: *Setup0* ($m = 1.15, leg = 3$), *Setup1* ($m = 0.75, leg = 1$), *Setup2* ($m = 1.25, leg = 0$), *Setup3* ($m = 0.85, leg = 0$), *Setup4* ($m = 1.0, leg = 2$)(20 trajectories per setup).

## 6 Conclusion

In this paper, we propose a decomposed mutual information optimization (DOMINO) framework to learn the generalized context for zero-shot dynamics generalization. The disentangled context is learned by maximizing the mutual information between the context and historical trajectories while minimizing the state transition prediction error. By decomposing the whole mutual information

Table 2: Comparison with *model-free* methods in more difficult multi-confounded environments (over 5 seeds).

| | Cripple-Ant-$m$-$d$ | | Cripple-Halfcheetah-$m$-$d$ | |
|---|---|---|---|---|
| | Train | Test | Train | Test |
| PEARL | 182±73 | 96 ±21 | **2538**±783 | 1028±445 |
| PPO+T-MCL | 187±65 | 109±36 | 2368±726 | 1006±434 |
| PPO+MINO | 206±64 | 113±34 | 2493±664 | 1197±424 |
| PPO+DOMINO | **233**±82 | **132**±27 | 2503±658 | **1326**±491 |

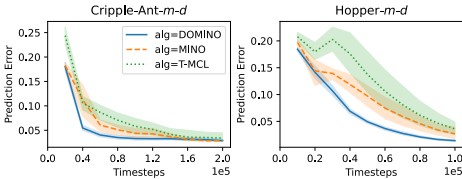

Figure 6: The testing prediction error.

optimization problem into smaller ones, DOMINO can reduce the need for samples collected in various environments and overcome the underestimation of the mutual information in multi-confounded environments. Extensive experiments illustrate that DOMINO benefits the generalization performance in unseen environments with both model-based RL and model-free RL. For future work, an effective combination of DOMINO and RIA [16], which expands the decomposed MI optimization to relational intervention approach proposed by RIA could become a stronger baseline for unsupervised dynamics generalization. We believe our work can lay the foundation of dynamics generalization in complex environments.

**Limitations and Negative Social Impact.** DOMINO sets the number of disentangled context vectors as a hyper-parameter equal to the number of confounders in the environments, and capturing the number of confounders automatically could be future works. We believe that DOMINO will not cause any negative social impact.

## Acknowledgments and Disclosure of Funding

The authors would like to thank the anonymous reviewers for their valuable comments and helpful suggestions. The work is supported by Huawei Noah's Ark Lab; Ping Luo is supported by the General Research Fund of HK No.27208720, No.17212120, and No.17200622.

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
