# Appendix of
# Decomposed Mutual Information Optimization for Generalized Context in Meta-Reinforcement Learning

**Yao Mu**
The University of Hong Kong
muyao@connect.hku.hk

**Yuzheng Zhuang**
Huawei Noah's Ark Lab
zhuangyuzheng@huawei.com

**Fei Ni**
Tianjin University
fei_ni@tju.edu.cn

**Bin Wang**
Huawei Noah's Ark Lab
wangbin158@huawei.com

**Jianyu Chen**
Tsinghua University
jianyuchen@tsinghua.edu.cn

**Jianye Hao**
Huawei Noah's Ark Lab
haojianye@huawei.com

**Ping Luo** *
The University of Hong Kong
pluo@cs.hku.hk

## A Derivations

### A.1 Proof of Lemma 1

According to the Barber and Agakov's variational lower bound [1], the mutual information $I(x; y)$ between $x$ and $y$ can be bounded as follows:

$$I(x; y) = \mathbb{E}_{p(x,y)} \log \frac{p(y|x)}{p(y)} \geq \mathbb{E}_{p(x,y)} \log \frac{q(y|x)}{p(y)}, \tag{1}$$

where $q$ is an arbitrary distribution. Specifically, $q(y|x)$ is defined by independently sampling a set of examples $\{y_1, \ldots, y_K\}$ from a proposal distribution $\pi(y)$ and then choosing $y$ from $\{y_1, \ldots, y_K\}$ in proportion to the importance weights $w_y = \frac{e^{\psi(x,y)}}{\sum_k e^{\psi(x,y_k)}}$, where $\psi$ is a function that takes $x$ and $y$ and outputs a scalar. According to the section 2.3 in [2], by setting the proposal distribution as the marginal distribution $\pi(y) \equiv p(y)$, the unnormalized density of $y$ given a specific set of samples $y_{2:K} = \{y_2, \ldots, y_K\}$ and $x$ is:

$$q\left(y \mid x, y_{2:K}\right) = p(y) \cdot \frac{K \cdot e^{\psi(x,y)}}{e^{\psi(x,y)} + \sum_{k=2}^{K} e^{\psi(x,y_k)}} \tag{2}$$

where $K$ denotes the numbers of samples. According to the equation 3 of section 2 in [3], the expectation of $q\left(y \mid x, y_{2:K}\right)$ with respect to resampling of the alternatives $y_{2:K}$ from $p(y)$ produces a normalized density:

$$\bar{q}(y \mid x) = \mathbb{E}_{p(y_{2:K})} \left[q\left(y \mid x, y_{2:K}\right)\right] \tag{3}$$

---

*Ping Luo is the corresponding author. Yao Mu and Fei Ni conducted this work during the internship in Huawei Noah's Ark Lab.

36th Conference on Neural Information Processing Systems (NeurIPS 2022).

With Equation 3 and Jensen's inequality applied in Equation 1, we have

$$I(x,y) \geq \mathbb{E}_{p(x,y)} \log \frac{\mathbb{E}_{p(y_{2:K})} q(y \mid x, y_{2:K})}{p(y)} \geq \mathbb{E}_{p(x,y)} \left[ \mathbb{E}_{p(y_{2:K})} \log \frac{q(y \mid x, y_{2:K})}{p(y)} \right]$$

$$= \mathbb{E}_{p(x,y)} \left[ \mathbb{E}_{p(y_{2:K})} \log \frac{p(y) K \cdot w_y}{p(y)} \right] \tag{4}$$

$$= \mathbb{E}_{p(x,y)} \left[ \mathbb{E}_{p(y_{2:K})} \log \frac{K \cdot e^{\psi(x,y)}}{e^{\psi(x,y)} + \sum_{k=2}^{K} e^{\psi(x,y_k)}} \right]$$

It is obviously that $\frac{e^{\psi(x,y)}}{e^{\psi(x,y)} + \sum_{k=2}^{K} e^{\psi(x,y_k)}} \leq 1$, thus we have

$$\mathbb{E}_{p(x,y)} \left[ \mathbb{E}_{p(y_{2:K})} \log \frac{K \cdot e^{\psi(x,y)}}{e^{\psi(x,y)} + \sum_{k=2}^{K} e^{\psi(x,y_k)}} \right] \leq \log K \tag{5}$$

With Equation 5, we have

$$\mathbb{E}_{p(x,y)} \left[ \mathbb{E}_{p(y_{2:K})} \log \frac{K \cdot e^{\psi(x,y)}}{e^{\psi(x,y)} + \sum_{k=2}^{K} e^{\psi(x,y_k)}} \right]$$

$$= \mathbb{E}_{p(x,y_1)p(y_{2:K})} \left[ \log \frac{e^{\psi(x,y)}}{\frac{1}{K} \sum_{k=1}^{K} e^{\psi(x,y_k)}} \right] = I_{\text{NCE}}(x;y \mid \psi, K) \leq \log K, \tag{6}$$

Therefore, we have

$$I(x,y) \geq I_{\text{NCE}}(x;y \mid \psi, K) \leq \log K \tag{7}$$

If $I(x,y) > \log K$, then $I(x,y) > \log K \geq I_{\text{NCE}}(x;y \mid \psi, K)$, and $I_{\text{NCE}}$ will be a loose bound.

Thus, $I_{\text{NCE}} \leq I(x,y) \leq \log K$ is the necessary condition for $I_{\text{NCE}}$ to be a tight bound of $I(x,y)$.

## A.2 Detailed derivation of Theorem 1

As the number of confounders increases, although the true mutual information $I(c; \mathcal{T})$ does not increase, the necessary condition of $I_{NCE}$ to be a tight lower bound of $I_{NCE}$ becomes more difficult to satisfy, and the demand of data increases significantly.

As for an entangled context, the necessary condition of the InfoNCE lower bound $I_{NCE}(c; \mathcal{T})$ to be a tight bound is

$$I_{NCE}(c; \mathcal{T}) \leq I(c; \mathcal{T}) \leq \log K \tag{8}$$

Since $I(c; \mathcal{T}) \geq \sum_{i=0}^{N} I(u_i; \mathcal{T})$, to let the above condition satisfied, the amount of data $K$ must satisfy

$$\log K \geq \sum_{i=0}^{N} I(u_i; \mathcal{T}) \tag{9}$$

$$K \geq e^{\sum_{i=0}^{N} I(u_i; \mathcal{T})} \tag{10}$$

Therefore, if the number of confounders increases, then the demand for data will grow exponentially.

When data is not rich enough, the nesseray condition may not be satisfied. The InfoNCE lower bound $I_{NCE}(c; \mathcal{T})$ may be loose, that is $I_{NCE}(c; \mathcal{T})$ may be much smaller than the true mutual information $I(c; \mathcal{T})$, thus the MI optimization based on $I_{NCE}(c; \mathcal{T})$ will be severely affected.

$I_{NCE}(c_i; \mathcal{T})$ is the lower bound of $I(c_i; \mathcal{T})$ and the necessary condition of $I_{NCE}(c_i; \mathcal{T})$ to be a tight bound of $I(c_i; \mathcal{T})$ is

$$I_{NCE}(c_i; \mathcal{T}) \leq I(c_i; \mathcal{T}) \leq \log K \tag{11}$$

As for disentangled context $c = \{c_1, c_2, \cdots, c_N\}$, we then derive the necessary condition of $I(c, \mathcal{T})$ to be a tight lower bound of $I(c, \mathcal{T})$:

With the assumption that the contexts $\{c_1, c_2, \cdots, c_N\}$ are independent to each other, then $I(c; \mathcal{T})$ could be derived as $\sum I(c_i; \mathcal{T})$. Therefore, under the confounder independent assumption, let $I_{NCE}(c; \mathcal{T})$ be a tight bound is only necessary to let every $I_{NCE}(c_i; \mathcal{T})$ to be a tight bound.

If every $I_{NCE}(c_i; \mathcal{T})(i = 1, 2, \ldots, N)$ is a tight bound, then we have

$$I_{NCE}(c_i; \mathcal{T}) \leq I(c_i; \mathcal{T}) \leq \log K \tag{12}$$

under the confounder independent assumption, we have

$$\sum I_{NCE}(c_i; \mathcal{T}) \leq \sum I(c_i; \mathcal{T}) \leq N \log K \tag{13}$$

$$I_{NCE}(c; \mathcal{T}) = \sum I_{NCE}(c_i; \mathcal{T}) \leq I(c; \mathcal{T}) = \sum I(c_i; \mathcal{T}) \leq N \log K \tag{14}$$

Thus, the necessary condition of $I_{NCE}(c; \mathcal{T})$ to be a tight bound of $I(c; \mathcal{T})$ could be relaxed to

$$I_{NCE}(c; \mathcal{T}) \leq I(c; \mathcal{T}) \leq N \log K \tag{15}$$

Therefore, by decomposing the MI estimation under the confounder independent assumption, the demand of the amount $K$ of data could be reduced from $K \geq e^{I(c;\mathcal{T})}$ to $K \geq e^{\frac{1}{N}I(c;\mathcal{T})}$. And with $I(c; \mathcal{T}) \geq \sum_{i=0}^{N} I(u_i; \mathcal{T})$, specifically, the the amount $K$ of data could be reduced from $K \geq e^{\sum_{i=0}^{N} I(u_i;\mathcal{T})}$ to $K \geq e^{\frac{1}{N}\sum_{i=0}^{N} I(u_i;\mathcal{T})}$.

## B  Pseudo-code

### B.1  Combination with model-based methods

We provide the pseudo-code of DOMINO combined with model-based methods. Firstly, the past state-action pairs are encoded into the disentangled context vectors by the context encoder. According to the learned context, the transition prediction network predicts the future states of different actions. Then, the context encoder is optimized by maximizing the mutual information between the disentangled context vectors and historical trajectories while minimizing the state transition prediction error. In particular, we use the cross entropy method (CEM) [4], a typical neural model predictive control (MPC) [5] method, to select actions, in which several candidate action sequences are iteratively sampled from a candidate distribution, which is adjusted based on best-performing action samples.

**Algorithm 1** Training DOMINO with context-aware world model

---

**Inputs**: learning rate $\alpha$, maximum number of iteration $P$, batch size $B$, the number of past observations $H_{\text{past}}$, maxium rollout step $max\_step$ and the number of future observations $H_{\text{future}}$.
Initialize parameters of prediction network $\phi$, context encoder $\varphi$.
Initialize replay buffer $\mathcal{D} \leftarrow \emptyset$.
**for** $P$ iterations **do**
    // COLLECT TRAINING SAMPLES
    $step = 0$
    $\mathcal{V} = 0$
    **while** $step \leq max\_step$ **do**
        Sample $u_{\mathcal{V}} \sim p_{u_{\text{train}}}(u)$.
        $\mathcal{V} = \mathcal{V} + 1$
        **for** $t = 1$ **to** TaskHorizon **do**
            $step = step + 1$
            Get context latent vectors $c_{t_0}, c_{t_1}, \ldots, c_{t_N} = g\left(\tau_t; \varphi\right), \tau_t = \{s_l, a_l\}_{l=t-H_{\text{past}}}^{t-1}$
            Collect samples $\{(s_t, a_t, s_{t+1}, r_t, \tau_t)\}$ from the environment using the planning algorithm based on CEM with the context vectors
        **end for**
        Update $\mathcal{D}_{u_{\mathcal{V}}} \leftarrow \mathcal{D}_{u_{\mathcal{V}}} \cup \{(s_t, a_t, s_{t+1}, r_t, \tau_t)\}$
    **end while**
    // UPDATE DYNAMICS MODELS AND ENCODER
    Initialize batch $\mathcal{B} \leftarrow \emptyset$.
    **for** $i = 1$ **to** $B$ **do**
        sample $\mathcal{V}^*$ from $[0, \mathcal{V}_{max}]$
        Sample $\{s_t, a_t, s_{t+1}, r_t, \tau_t\}$ from $\mathcal{D}_{u_{\mathcal{V}^*}}$
        Sample positive trajectories $\tau^+$ from $\mathcal{D}_{u_{\mathcal{V}^*}}$
        Sample negative trajectories $\left\{\tau_k^-\right\}_{k=2}^{K}$ from $\mathcal{D}_{u_{\mathcal{V}! = \mathcal{V}^*}}$
        Get context latent vectors $c_{t_0}, c_{t_1}, \ldots, c_{t_N} = g\left(\tau_t; \varphi\right)$
        Update $\mathcal{B} \leftarrow \mathcal{B} \cup \{(s_t, a_t, s_{t+1}, r_t, \tau_t)\}$
    **end for**
    $\mathcal{L}^{\text{pred}} \leftarrow E_{\tau^* \sim \mathcal{B}}\left[-\frac{1}{H}\sum_{\lambda=t}^{t+H_{\text{future}}-1} \log f_\phi\left(s_{\lambda+1} \mid s_\lambda, a_\lambda, (c_{0_\lambda}, \ldots, c_{N_\lambda})\right)\right]$
    $\mathcal{L}^{\text{NCE}} \leftarrow \sum_i^N I_{\text{NCE}}\left(c_i; \mathcal{T}\right) - \sum_j^N \sum_{i=0, i \neq j}^N I_{\text{NCE}}\left(c_i; c_j\right)$
    Update $\varphi \leftarrow \varphi - \alpha \nabla_\varphi \mathcal{L}^{\text{NCE}}$
    Update $\varphi \leftarrow \varphi - \alpha \nabla_\varphi \mathcal{L}^{\text{pred}}$
    Update $\phi \leftarrow \phi - \alpha \nabla_\phi \mathcal{L}^{\text{pred}}$
**end for**

---

## B.2 Combination with model-free methods

We provide the pseudo-code for the combination between DOMINO and the model-free method, which uses the context encoder learned by DOMINO as a plug-and-play module to extract accurate context. We concatenate the disentangled context encoded by a pre-trained context encoder from DOMINO and the current state-action pairs, and learn a conditional policy $\pi\left(a_t | s_t, c_0, \ldots, c_N\right)$. We choose the Proximal Policy Optimization (PPO) method [6] to train the agents.

## C Details about the testing environments

```python
def change_env(self):
        mass = np.copy(self.original_mass)
        damping = np.copy(self.original_damping)
        mass *= self.mass_scale
        damping *= self.damping_scale
        self.model.body_mass[:] = mass
        self.model.dof_damping[:] = damping
```

Listing 1: PyTorch-style pseudo-code for dynamics change based on Mujoco engine.

**Algorithm 2** Proximal Policy Optimization with disentangled context encoder learned by DOMINO

**Inputs**:Maximum number of iteration $P$, number of actor updates $M$, number of critic updates $B$, the KL regular coefficient $\lambda$, scaling coefficient $\alpha$ and the learning rate $\beta$.
Initialize parameters of policy network $\theta$, value network $\xi$, and context encoder $\varphi$.
**for** $P$ iterations **do**
    Encode disentangled context vectors $c_{t_0}, c_{t_1}, \ldots, c_{t_N}$ by the learned context encoder $g_\varphi(\cdot)$
    Run policy $\pi_\theta$ for $T$ timesteps, collecting $\{\{s_t, c_{t_0}, c_{t_1}, \ldots, c_{t_N}, a_t, r_t\}\}_{t=1}^T$
    Estimate advantages $\hat{A}_t = \sum_{t' > t} \gamma^{t'-t} r_{t'} - V_\xi(s_t, c_{t_0}, c_{t_1}, \ldots, c_{t_N})$
    $\pi_{\text{old}} \leftarrow \pi_\theta$
    **for** $M$ updates **do**
        $J_{\text{PPO}}(\theta) \leftarrow - \left\{ \sum_{t=1}^T \frac{\pi_\theta(a_t|s_t, c_{t_0}, c_{t_1}, \ldots, c_{t_N})}{\pi_{old}(a_t|s_t, c_{t_0}, c_{t_1}, \ldots, c_{t_N})} \hat{A}_t - \lambda \text{KL}[\pi_{old}|\pi_\theta] \right\}$
        $\theta \leftarrow \theta - \beta \nabla_\theta J_{\text{PPO}}$
    **end for**
    **for** $B$ updates **do**
        $L_{\text{BL}}(\xi) \leftarrow \sum_{t=1}^T (\sum_{t' > t} \gamma^{t'-t} r_{t'} - V_\xi(s_t, c_{t_0}, c_{t_1}, \ldots, c_{t_N}))^2$
        $\xi \leftarrow \xi - \beta \nabla_\xi L_{\text{BL}}$
    **end for**
    **if** $\text{KL}[\pi_{old}|\pi_\theta] > \beta_{\text{high}} \text{KL}_{\text{target}}$ **then**
        $\lambda \leftarrow \alpha\lambda$
    **else if** $\text{KL}[\pi_{old}|\pi_\theta] < \beta_{\text{low}} \text{KL}_{\text{target}}$ **then**
        $\lambda \leftarrow \lambda/\alpha$
    **end if**
**end for**

```python
def reset_model(self):
        c = 0.01
        self.set_state(
            self.init_qpos + self.np_random.uniform(low=-c, high=c, size=self.model.
    nq),
            self.init_qvel + self.np_random.uniform(low=-c, high=c, size=self.model.
    nv,)
        )
        pos_before = mass_center(self.model, self.sim)
        self.prev_pos = np.copy(pos_before)

        random_index = self.np_random.randint(len(self.mass_scale_set))
        self.mass_scale = self.mass_scale_set[random_index]

        random_index = self.np_random.randint(len(self.damping_scale_set))
        self.damping_scale = self.damping_scale_set[random_index]

        self.change_env()
        return self._get_obs()
```

Listing 2: PyTorch-style pseudo-code for multi-confounded environments initialization.

For CartPole environments, we use open-source implementation of CartPoleSwingUp-v2[2], which is the modified version of original CartPole environments from OpenAI Gym. The objective of CartPole task is to swing up the pole by moving a cart and keep the pole upright. For our experiments, we modify the push force $f$ and the pole length $l$ simultaneously. As for Pendulum, we scale the pendulum mass by scale factor $m$ and modify the pendulum length $l$. For Pendulum environments, we use the open-source implementation of from the OpenAI Gym. The objective of Pendulum is to swing up the pole and keep the pole upright within 200 timesteps. We scale the pendulum mass by scale factor $m$ and modify the pendulum length $l$.

As for Hopper, Half-cheetah, Ant, and Slimhumanoid, we use the environments from MuJoCo physics engine [3], and scale the mass of every rigid link by scale factor $m$, and scale damping of every joint by scale factor $d$. As for Crippled Ant and Crippled Half-cheetah, we scale the mass of every rigid link by scale factor $m$, scale damping of every joint by scale factor $d$, and randomly

---

[2]We use implementation available at https://github.com/0xangelo/gym-cartpole-swingup

[3]We use implementation available at https://github.com/iclavera/learning_to_adapt

Table 3: Environment parameters used for the multi-confounded experiments.

| | | Train | | Test |
|---|---|---|---|---|
| CartPole | $f \in$ | $\{5.0, 6.0, 7.0, 8.0, 9.0, 10.0,$ $11.0, 12.0, 13.0, 14.0, 15.0\}$ | $f \in$ | $\{3.0, 3.5, 16.5, 17.0\}$ |
| | $l \in$ | $\{0.40, 0.45, 0.50, 0.55, 0.60\}$ | $l \in$ | $\{0.25, 0.30, 0.70, 0.75\}$ |
| Pendulum | $m \in$ | $\{0.75, 0.80, 0.85, 0.90, 0.95,$ $1.0, 1.05, 1.10, 1.15, 1.20, 1.25\}$ | $m \in$ | $\{0.50, 0.70, 1.30, 1.50\}$ |
| | $l \in$ | $\{0.75, 0.80, 0.85, 0.90, 0.95,$ $1.0, 1.05, 1.10, 1.15, 1.20, 1.25\}$ | $l \in$ | $\{0.50, 0.70, 1.30, 1.50\}$ |
| Half-cheetah | $m \in$ | $\{0.75, 0.85, 1.0, 1.15, 1.25\}$ | $m \in$ | $\{0.40, 0.50, 1.50, 1.60\}$ |
| | $d \in$ | $\{0.75, 0.85, 1.0, 1.15, 1.25\}$ | $d \in$ | $\{0.40, 0.50, 1.50, 1.60\}$ |
| Ant | $m \in$ | $\{0.75, 0.85, 1.0, 1.15, 1.25\}$ | $m \in$ | $\{0.40, 0.50, 1.50, 1.60\}$ |
| | $d \in$ | $\{0.75, 0.85, 1.0, 1.15, 1.25\}$ | $d \in$ | $\{0.40, 0.50, 1.50, 1.60\}$ |
| SlimHumanoid | $m \in$ | $\{0.80, 0.90, 1.0, 1.15, 1.25\}$ | $m \in$ | $\{0.60, 0.70, 1.50, 1.60\}$ |
| | $d \in$ | $\{0.80, 0.90, 1.0, 1.15, 1.25\}$ | $d \in$ | $\{0.60, 0.70, 1.50, 1.60\}$ |
| Crippled Ant | $m \in$ | $\{0.75, 0.85, 1.0, 1.15, 1.25\}$ | $m \in$ | $\{0.40, 0.50, 1.50, 1.60\}$ |
| | $d \in$ | $\{0.75, 0.85, 1.0, 1.15, 1.25\}$ | $d \in$ | $\{0.40, 0.50, 1.50, 1.60\}$ |
| | crippled leg:$\{0, 1, 2\}$ | | crippled leg:$\{3\}$ | |
| Crippled Halfcheetah | $m \in$ | $\{0.75, 0.85, 1.0, 1.15, 1.25\}$ | $m \in$ | $\{0.40, 0.50, 1.50, 1.60\}$ |
| | $d \in$ | $\{0.75, 0.85, 1.0, 1.15, 1.25\}$ | $d \in$ | $\{0.40, 0.50, 1.50, 1.60\}$ |
| | crippled leg:$\{0\}$ | | crippled leg:$\{1\}$ | |

select one leg, and make it crippled. The objectives of these tasks are to move forward as fast as possible while minimizing the action cost. The detailed settings are illustrated in Table 3. We provide the pyTorch-style pseudo-code for multi-confounded environments in Listing 1 and Listing 2. We implement these environments based on the publicly available code provide by [7, 8], and we also open-source the code of the multiple-confounded environments[4]. For both the training and testing phase, we sample the confounders at the beginning of each episode. During training, we randomly select a combination of confounders from a training set. At test time, we evaluate each algorithm in unseen environments with confounders outside the training range. We also provide the PyTorch-style pseudo-code for the dynamics change based on Mujoco engine.

# D  Implementation details

## D.1  Combination with Model-based RL

The context encoder is modeled as multi-layer perceptrons (MLPs) with 3 hidden layers and N output heads which are single-layer MLPs. Every disentangled context vector is produced as a 10-dimensional vector by the 3 hidden layers and a specific output head. Then, the disentangled context vectors are used as the additional input to the prediction network, i.e., the input is given as a concatenation of state, action, and context vector. We use $H_{past} = 10$ for the number of past observations and $H_{future} = 5$ for the number of future observations. The prediction network is modeled as multi-layer perceptrons (MLPs) with 4 hidden layers of 200 units each and Swish activations. For each prediction head, the mean and variance are parameterized by a single linear layer that takes the output vector of the backbone network as an input. To train the prediction network, we collect 10 trajectories with 200 timesteps from environments using the MPC controller and train the model for 50 epochs at every iteration. We train the prediction network for 10 iterations for every experiment. We evaluate trained models on environments over 8 random seeds every iteration to report the testing performance. The Adam optimizer [9] is used with a learning rate $1 \times 10^{-4}$. For

---

[4] We provide open-source environments at https://anonymous.4open.science/r/Multiple-confounded-Mujoco-Envs-01F3

planning, we use the cross entropy method (CEM) with 200 candidate actions for all the environments. The horizon of MPC is set as 30.

## D.2 Combination with Model-free RL

We train the model-free agents for 5 million timesteps on OpenAI-Gym and MuJoCo environments (i.e., Hopper, Half-cheetah, Ant, Crippled Half-cheetah, Crippled Half-Ant, Slim-Humanoid) and 0.5 million timesteps on CartPole and Pendulum. The trained agents are evaluated every 10,000 timesteps over 5 random seeds. We use a discount factor $\gamma = 0.99$, a generalized advantage estimator [10] parameter $\lambda = 0.95$ and an entropy bonus of 0.01 for exploration. In every iteration, the agent rollouts 200 timesteps in the environments with the learned policy, and then it will be trained for 8 epochs with 4 mini-batches. The Adam optimizer is used with the learning rate $5 \times 10^{-4}$.

## D.3 Details of InfoNCE

We provide detailed pseudocode for the calculation of InfoNCE bound. Specifically, the temperature $\tau$ is set as 0.004 to the calculation of $I(c_i, \mathcal{T})$ and is set as 0.1 to calculation of $I(c_i, c_j)$.

---

**Algorithm 3** Pseudocode of InfoNCE in a PyTorch-like style.

```
# x_q input vector
# x_k positive sample
# x_que negative samples
# f_q, f_k, f_que: encoder networks for query, key and queue
# m: momentum
# t: temperature

def InfoNCE(x_q,x_k,x_que):
    q = f_q.forward(x_q) # queries: NxC
    k = f_k.forward(x_k) # keys: NxC
    k = k.detach() # no gradient to keys
    queue = f_que.forward(x_que) # keys: Cx(K-1)

    # positive logits: Nx1
    l_pos = bmm(q.view(N,1,C), k.view(N,C,1))

    # negative logits: Nx(K-1)
    l_neg = mm(q.view(N,C), queue.view(C,K-1))

    # logits: NxK
    logits = cat([l_pos, l_neg], dim=1)

    # contrastive loss, Eqn.(1)
    labels = zeros(N) # positives are the 0-th
    loss = CrossEntropyLoss(logits/t, labels)
```

---

## D.4 Details of the adaptive planning used in adaption process

The prediction model has 3 output head $head_0, head_1, head_2$ which are used for selecting actions by planning. The adaptive planning method selects the most accurate prediction head over a recent experience. Given $N$ past transitions, we select the prediction head $h_*$ by

$$\underset{head \in [Head]}{\operatorname{argmin}} \sum_{i=t-N}^{t-2} \ell\left(s_{i+1}, f\left(s_{i+1} \mid s_i, a_i, (c_{0_\lambda}, \ldots, c_{N_\lambda}); \phi, head\right)\right)$$

where $\ell$ is the mean square error function. All the hyper-parameter is set as same as T-MCL[11].

# E  Additional Results

## E.1 Prediction Error

As shown in Figure 1, DOMINO has a smaller prediction error compared to T-MCL and its ablation version MINO (optimize MI with entangled context), indicating that the learned context can effectively help predict the future state more accurately, which is the key to the performance of the model-based planning.

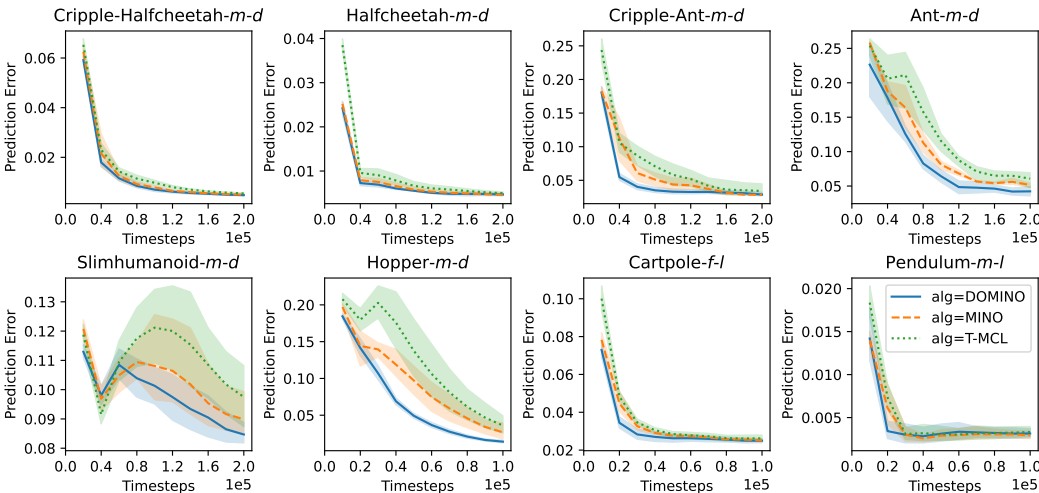

Figure 1: Comparison with the *model-based methods* of the Prediction Error. The results show the mean and standard deviation of average returns averaged over 8 runs.

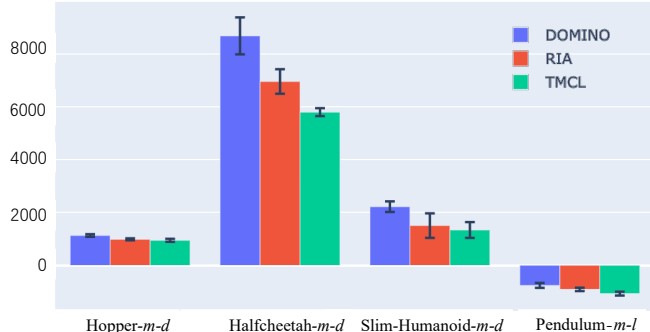

Figure 2: Generalization performance comparison between DOMINO, RIA and TMCL over 5 runs (DOMINO and T-MCL are with adaptive planning).

### E.2  More results on the comparison with RIA

We provide the comparison between the DOMINO without adaptive planning and RIA in the main paper. Here, we compare DOMINO and T-MCL with adaptive planning with RIA under multi-confounded setting, the environments including Hopper-$m$-$d$, Halfcheetah-$m$-$d$, Slim-humanoid-$m$-$d$ and Pendulum-$m$-$l$. As shown in Figure 2, DOMINO also achieves better generalization performance than RIA and the TMCL with adaptive planning.

### E.3  Sensitivity Analysis of the hyper-parameter N

We compare the performance of DOMINO with different hyper-parameter $N$, which is equal or not equal to the number of confounders in the environment. In this experiment, the confounder is the damping, mass, and a crippled leg (number of confounders is 3), and we compare the performance of DOMINO with different hyper-parameter $N = 1, 2, 3, 4$. As shown in Figure 3, even though the hyper-parameter $N$ is not equal to the ground truth value of the confounder number, DOMINO also benefits the context learning compared to the baselines like TMCL.

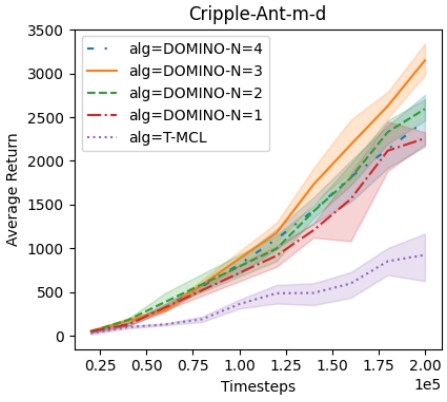

(a) Performance evaluation in seen environments

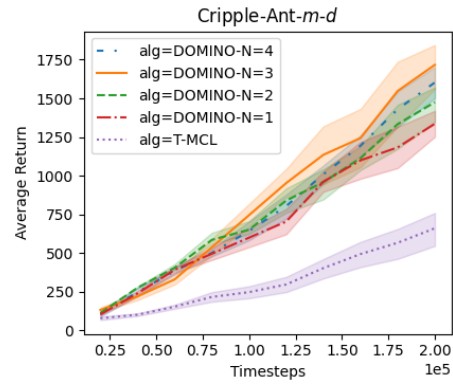

(b) Performance evaluation in unseen environments

Figure 3: The ablation of different N in Crippled-Ant-m-d domain(contains 3 confounders).

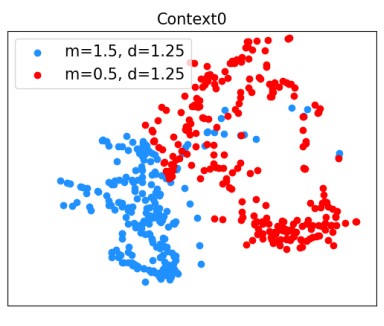

(a) Visualization of Context0

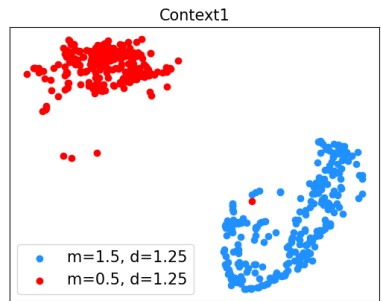

(b) Visualization of Context1

Figure 4: Visualization of disentangled context in with same damping scale $d = 1.25$ and different mass scale $m = 1.5, m = 0.5$.

## F    Visualization

### F.1    Verifying whether the contexts is disentangled

We add an additional experiment to show that the context vectors inferred by DOMINO are disentangled well. We vary only one of the confounders and observe the changes of $N$ disentangled vectors. In this experiment, we set up two different confounders: mass $m$ and damping $d$. Under the DOMINO framework, the context encoder inferred two disentangled context vectors: context 0 and context 1. As shown in Figure 5 and Figure 4, the context 1 is more related to damping. When the confounders are set as the same mass but different damping, the visualization result of context 1 under different settings are separated clearly from each other, while under the same damping but different mass settings, the visualization result of context 1 is much more blurred from each other. Similarly, context 0 is more related to mass. When the confounders are set to the same damping but different mass, the visualization result of context 0 under different settings is separated clearly from each other, while under the same mass but different damping settings, the visualization result of context 0 is less different from each other.

### F.2    Visualization of the whole context

**Visualization**.    We visualize the whole context which is a the concatenation of the disentangled contexts learned by DOMINO via t-SNE [12] and compare it with the entangled context learned

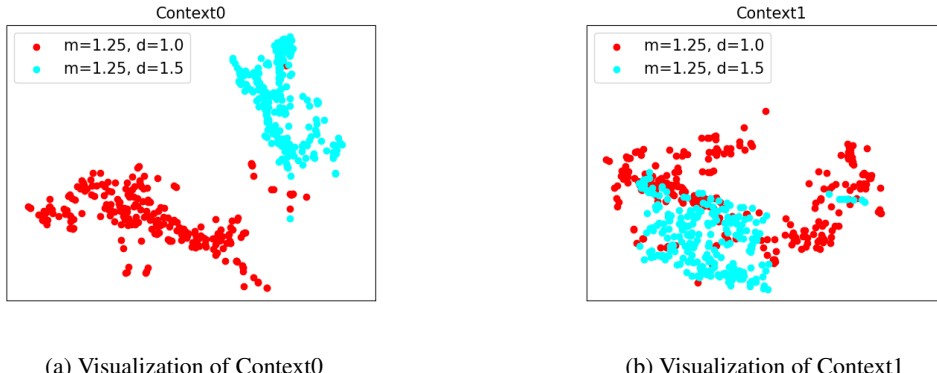

(a) Visualization of Context0          (b) Visualization of Context1

Figure 5: Visualization of disentangled context in with same mass scale $m = 1.25$ and different damping scale $d = 1.0, d = 1.5$.

by T-MCL. We run the learned policies under 5 randomly sampled setups of multiple confounders and collect 200 trajectories for each setting. Further, we encode the collected trajectories into context in embedding space and visualize via t-SNE [12] and PCA [13]. As shown in Figure 6 and Figure 7, we find that the disentangled context vectors encoded from trajectories collected under different confounder settings could be more clearly distinguished in the embedding space than the entangled context learned by T-MCL. This indicates that DOMINO extracts high-quality task-specific information from the environment compared with T-MCL. Accordingly, the policy conditioned on the disentangled context is more likely to get a higher expected return on dynamics generalization tasks, which is consistent with our prior empirical findings.

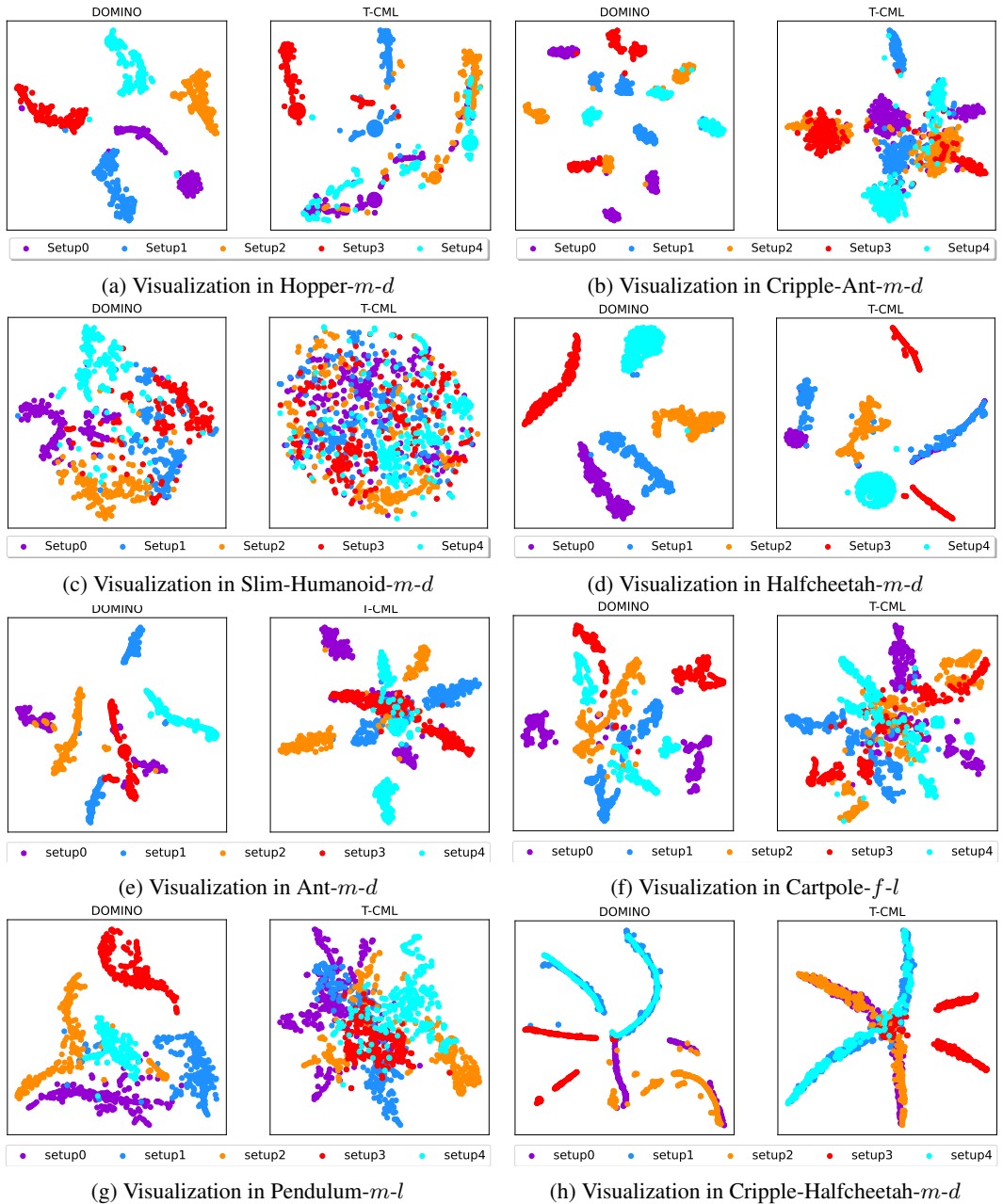

Figure 6: t-SNE [12] visualization of context vectors extracted from trajectories collected in various environments. Embedded points from environments with the same confounders have the same color.

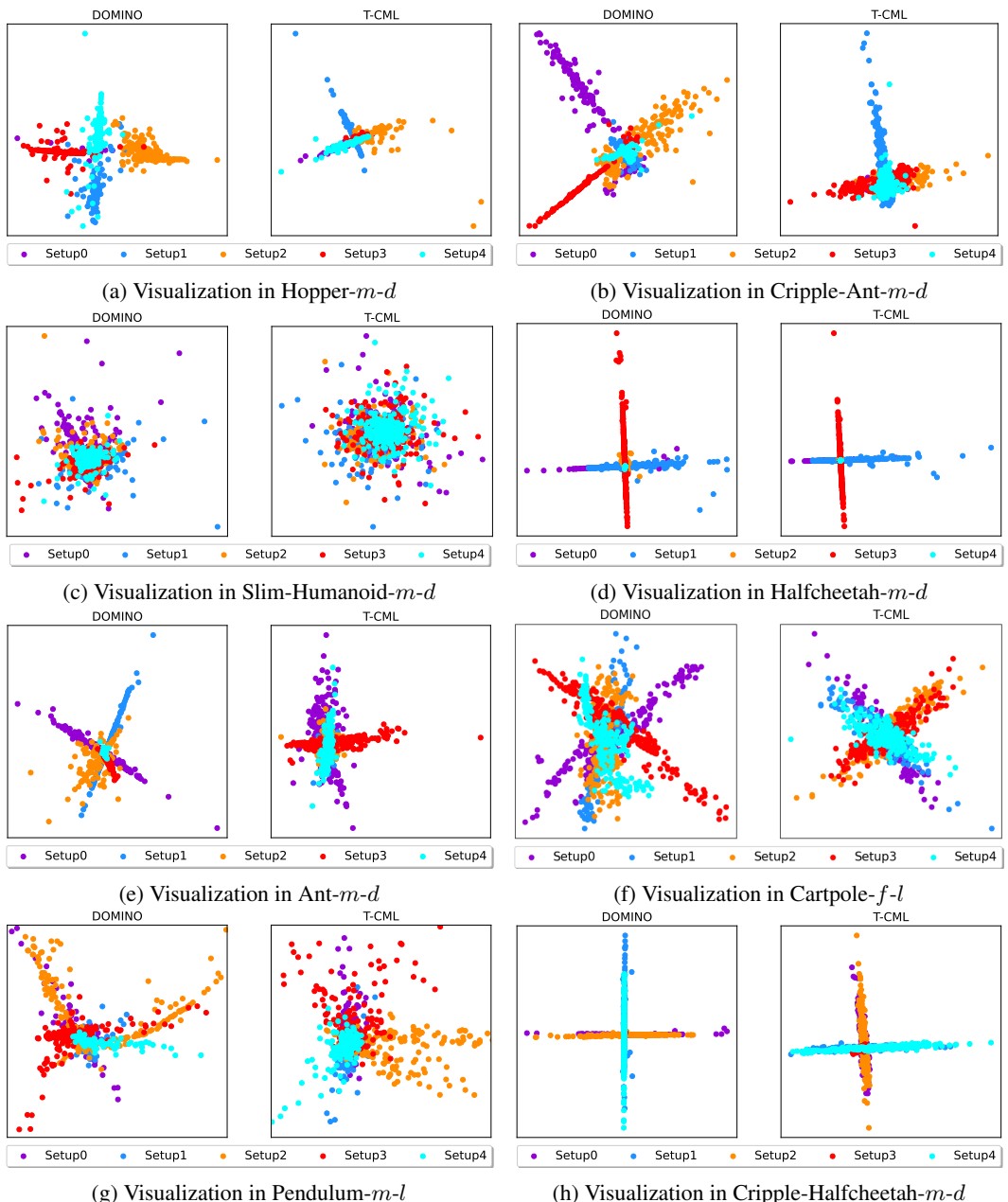

(a) Visualization in Hopper-$m$-$d$

(b) Visualization in Cripple-Ant-$m$-$d$

(c) Visualization in Slim-Humanoid-$m$-$d$

(d) Visualization in Halfcheetah-$m$-$d$

(e) Visualization in Ant-$m$-$d$

(f) Visualization in Cartpole-$f$-$l$

(g) Visualization in Pendulum-$m$-$l$

(h) Visualization in Cripple-Halfcheetah-$m$-$d$

Figure 7: PCA [12] visualization of context vectors extracted from trajectories collected in various environments. Embedded points from environments with the same confounders have the same color.

# G    Further discussion about the future works

## G.1    Expand DOMINO into reward generalization

The reward generalization can be categorized as a kind of task generalization. The parameter of the reward function, for example, the target speed of the robot, can also be considered as a confounder that influences the reward transition. To address this problem under the DOMINO framework, we provide the following solution. The context encoder maps the current sequence of state-action-reward pairs $\{s_\tau, a_\tau, r_\tau\}_{t-H}^{t}$ into disentangled contexts, which contains the information of the physical confounders like mass and damping and the reward confounder. The historical trajectory also should consider the reward part, i.e., $s_t, a_t, r_t, s_{t+1}$. Then the proposed decomposed mutual information optimization method can also be used in this situation to extract effective context. Moreover, the prediction loss should also add the reward prediction term. Thus, with the above design, DOMINO can address the reward generalization and dynamics generalization simultaneously.

## G.2    Expand DOMINO to support related confounders

To further support the complex environment with confounders related to each other, we can explore how to extract the information that is most useful for state transfer from each of the confounders separately when they do have some correlation with each other. One possible option is to adjust the penalty factor for mutual information between the context vectors in DOMINO, which can be set to be dynamically adjustable.

## G.3    Combined with VariBad and RIA

VariBad [14] introduces the VAE method and recurrent network to learn the context, which optimizes the context learning from different perspectives from DOMINO and TMCL methods. We believe the effective combination of DOMINO and Varibad will become a more powerful baseline for meta-RL. RIA[15] doesn't need to record if the two trajectories are collected in the same episode, since the relational intervention approach could optimize the mutual information without environment labels and even without the environment ID, which provides a promising direction of unsupervised dynamics generalization. We believe that DOMINO and RIA are not in competition, on the contrary, their effective combination will become a stronger baseline, for example, the decomposed MI optimization can be expanded into the relational intervention approach proposed in RIA.