# OpenReview forum: "DOMINO: Decomposed Mutual Information Optimization for Generalized Context in Meta-Reinforcement Learning"
_NeurIPS.cc/2022/Conference — NeurIPS 2022 Accept_

### Official Review · Reviewer_RkbU · 2022-07-07

**Rating:** 6
**Confidence:** 5
**Soundness:** 2 fair
**Presentation:** 3 good
**Contribution:** 2 fair

**Summary:**

This paper proposes a decomposed mutual information method to learn disentangled context information, which can generalize reinforcement learning algorithms into unseen environments. The experimental experiments demonstrate that the proposed method can achieve better performance than the previous methods.

**Questions:**

Please refer to the "Weakness" listed above.

**Limitations:**

Please refer to the "Weakness" listed above.

**Strengths And Weaknesses:**

Strengths:
1. The writing of this paper is pretty well, and the idea of it is easy to follow.
2. The figures in this paper are very clear and very well.
3. The extensive experiments show the effectiveness of the proposed method.

Weakness:
1. Based on the title, I assume that this study focuses on the meta-reinforcement learning problem. The conventional meta-reinforcement learning methods include an adaptation process, but this paper makes no mention of this process. Additionally, the paper states that it intends to train a general context-encoder to solve the adaptation problem, indicating that the paper's context is the dynamics generalization in reinforcement learning (this paper also mentions it in line 84), which is in contrast to the title of the paper, which refers to meta-reinforcement learning.

2. The second problem of this paper is the novelty. The paper aims to maximize the mutual information between contexts extracted from historical information and the historical trajectories. However, this paper does not make clear the relationship with [1,2,3] which also attempt to maximize the MI between context vector and historical trajectories. Furthermore, this work does not compare the performance with [3] and even does not acknowledge it, despite the fact that [3] focuses on a similar problem to this paper. As a result of the missing contribution and experimental comparisons with [1,2,3], I believe this paper's uniqueness is somewhat limited.

3. The number of learned context vectors $c$ is set as the number of environments in the study, which is the primary hyperparameter of the suggested technique. However, in a real-world setting, the number of environments is not available, making it unfair to compare it to the baseline TMCL, which doesn't rely on such prior information. This increases my concerns about the technical soundness of this paper.

In conclusion, while the writing and experimental results are excellent, this paper suffers from the aforementioned clarity and novelty issues. If the authors address my concerns in their response, I will consider raising my score.


------------------------------------------------- After Rebuttal ------------------------------------------------

I think that the additional experimental results and discussion in the revision resolve my concerns about the clarity problem of the submission, so I increase my score from 4 to 6 accordingly.

Minors: I believe that RIA considers context information and constructs confounder sets with multiple confounders, so I believe that RIA should be discussed in the introduction's confounder discussion (Line 42).

[1] Haotian Fu, Hongyao Tang, Jianye Hao, Chen Chen, Xidong Feng, Dong Li, and Wulong Liu. Towards effective context for meta-reinforcement learning: an approach based on contrastive learning.

[2] Li, L., Huang, Y., Chen, M., Luo, S., Luo, D., & Huang, J. (2021). Provably Improved Context-Based Offline Meta-RL with Attention and Contrastive Learning. arXiv preprint arXiv:2102.10774.

[3] Guo J, Gong M, Tao D. A Relational Intervention Approach for Unsupervised Dynamics Generalization in Model-Based Reinforcement Learning[C]//International Conference on Learning Representations. 2022.

---

> ### Author Response · Authors · 2022-08-02
> **Response to Reviewer RkbU (Part 2/2)**
>
>
>
> **Q3. The number of learned context vectors as prior information to compare with TMCL.**
>
> ANS:
>
> Good question!  The number of learned context vectors is set as the number of confounders in the environment as a primary hyper-parameter.
>
> Here, we add additional sensitivity test experiments of the hyper-parameter $N$ to solve your concerns.
>
> We compare the performance of DOMINO with different hyper-parameter $N$, which is equal or not equal to the number of confounders in the environment. In this experiment, the confounder is the damping, mass, and a crippled leg (number of confounders is 3), and we compare the performance of DOMINO with different hyper-parameter $N={1,2,3}$. As shown in Figure 9 (**see Appendix E.3, page 21, revised version paper**), even though the hyper-parameter $N$ is not equal to the ground truth value of the confounder number, DOMINO also benefits the context learning. In practice, under a conservative setup of hyper-parameter $N$, DOMINO can also benefit the context learning compared to the baselines like TMCL.
>
> We acknowledge that the introduction of the prior information is one of the limitations of our paper, and we also explicitly state this in the limitation section. Here, we provide a practical method to estimate the number of confounders. In the absence of a prior, one can consider training multiple context encoders in parallel, and selecting the best $N$ by comparing the accuracy of state transition prediction, etc.
>
> This paper focuses on verifying that decomposed mutual information optimization has a significant advantage over entangled mutual information optimization when multiple confounders act together. And we will continue to explore the methods without the prior information on confounder numbers in future works.

---

> > ### Comment · Reviewer_RkbU · 2022-08-06
> > **Response to Authors**
> >
> > I appreciate the author's response and point-by-point response. I believe that some of my concerns have been addressed by the author's response, but I do still have the following concerns:
> >
> > Response 1: I believe that the revised version's expanded explanation of the adaptation process clarifies the submission's setting.
> >
> > Response 2: My concern about the paper's clarity, in my opinion, has not been fully addressed by the response, and my main concern is how the submission and RIA are related. The following are the explanations:
> >
> >   a) The experimental setting between RIA and the submission is very relevant: CCM[1] and Focal[2] concentrate on the setting of a single confounder, but RIA[3] also performs experiments in a situation with numerous confounders as the submission does.
> >
> >   b) The method of RIA is relevant to the submission: RIA also views context information as a confounder between $s t$ and $s_t+1$.  To extract disentangled context information in the multiple confounders setting. RIA uses MI and additional interventional to extract disentangled context information.
> >
> > I, therefore, think that RIA is quite relevant to the submission regardless of the experimental setting and method, and so the differences/advantages/experimental/visualization comparisons between the submission and RIA  should be made clearer in the Introduction/Related Work and Experiment of the main paper so that the contribution and significance of the submission can be better clarified.
> > Furthermore, RIA only uses one prediction head as their codes show, so it does not add the intervention module to TMCL as this submission does (as the description in the Relevant work in the revised submission).
> >
> > Response 3:  I appreciate the author's further experimental sensitive analysis. The analysis demonstrates that when $N$ is less than the actual number of confounders, the suggested method is not sensitive to $N$. In some circumstances, however, $N$ might be higher than the actual number of confounders. Can the authors run additional tests to verify that the suggested approach is reliable when $N$ is greater than the actual number of confounders?
> >
> > Open Discussion:
> > The submission makes the straightforward and logical assumption that the confounders are independent and seeks to extract disentangled context information from multiple confounder settings. To extract disentangled context information from historical transitions, however, should be an ill-posed problem because even if the confounders are independent, the transition functions that map from $s_t,a_t$ to $s_{t+1}$ are not identifiable in the presence of multiple confounders. I suspect that disentangled context information cannot be extracted by only optimizing the MI objective function.
> >
> > Some further questions: I notice that the positive trajectories in this paper are collected in the same setting as the confounders (Line 184), emphasizing the fact that the environment ID is provided in the submission. However, because the confounder value can sometimes be difficult to label, TMCL/RIA assume that the detailed setting of the confounder is not available. As a result, these two methods learn the context encoder unsupervised. Therefore, I am not sure whether the experimental comparisons are fair or not.
> >
> >
> >
> >
> > Minors:
> > RIA is published in ICLR 2022, not 2021.

---

> > > ### Author Response · Authors · 2022-08-09
> > > **Response to your further concerns (Part2/2 [Open Discussion])**
> > >
> > > **Q4: whether the disentangled context information can be extracted by only optimizing the MI objective function?**
> > >
> > > ANS:
> > >
> > > DOMINO learns the disentangled context not only by optimizing the MI objective function.
> > >
> > > To learn the disentangled context vectors, DOMINO optimizes the mutual information between each disentangled context vector and historical trajectory separately while regularizing the mutual information between the disentangled context vectors each other(see Equation 5). Furthermore, the state transition prediction loss will also help to learn the context encoder(see Equation 8 and Equation 9).
> > >
> > > **Q5: About whether the detailed setting of the confounder is used in DOMINO.**
> > >
> > > ANS:
> > >
> > > Good question. DOMINO doesn't use the specific confounder value.
> > > Since the environment is randomly initialized by specifying the combination of confounders at the beginning of each episode, and confounders remain unchanged until the end of the episode(as same as T-MCL, CADM, and RIA), DOMINO considers the trajectories collected in the same episode as positive examples and consider the trajectories generated in other episodes as a negative example. DOMINO treats negative cases equally and does not needs the specific label of the value of each confounder.
> > > We appreciate the relational intervention approach proposed in RIA, we believe that the combination of DOMINO and RIA can be a more sample-efficient baseline without the dependency on any environment label.
> > >
> > >
> > > Thanks again for reading our article carefully and giving very constructive suggestions. We hope we resolve all of your concerns and we wish you could reconsider your score.

---

> > > > ### Comment · Reviewer_RkbU · 2022-08-09
> > > > **Response to Authors' Further Revision**
> > > >
> > > > Thanks for the authors' further response and effort. I think that the additional experimental results and discussion in the revision resolve my concerns about the clarity problem of the submission, so I increase my score from 4 to 6 accordingly.
> > > >
> > > > Minors: I believe that RIA considers context information and constructs confounder sets with multiple confounders, so I believe that RIA should be discussed in the introduction's confounder discussion (Line 42).

---

> > > > > ### Author Response · Authors · 2022-08-09
> > > > > **Sincerely thank you for your recognition of our work**
> > > > >
> > > > > We sincerely thank you for your detailed and constructive suggestions, and we have revised the relevant parts of the introduction following your kind suggestion.
> > > > >
> > > > > With your help we have revised the paper and improved the quality of our paper. We really appreciate your effort to review our paper and recognition of our work. Thanks a lot.

---

> > > ### Author Response · Authors · 2022-08-09
> > > **Response to your further concerns (Part 1/2)**
> > >
> > > Thanks for your kind suggestion.
> > >
> > > We added the discussion of the relationship between DOMINO and RIA  in the introduction, and related works, and list RIA as a key baseline in experiments in the third revised paper (mark as purple).
> > >
> > >
> > > **Q1:The relationship between DOMINO and RIA**
> > >
> > > ANS:
> > >
> > > We carefully explain the relationship between DOMINO and RIA as follows:
> > >
> > > Firstly, the key differences between RIA and DOMINO are:
> > >
> > > 1) DOMINO infers several disentangled context vectors concurrently from the current sequence of state action pairs, while  RIA infers a centralized context vector to represent the environmental characteristic from the current sequence of state action pairs.
> > >
> > > 2) DOMINO divides the mutual information problem into the summation of several smaller ones to reduce the demand for data, which optimizes the mutual information between every disentangled context vector and historical trajectory separately and regularizes the mutual information between the disentangled context vectors, while RIA directly optimizes the whole mutual information between the inferred context vector and historical trajectory, which may also suffer from the problem that the optimize objective may be a loose bound of the true mutual information with not enough data when multiple factors that affect the state transition concurrently.
> > >
> > > 3)  DOMINO considers the trajectories collected in the same episode as positive examples and considers the trajectories generated in other episodes as negative examples, since the environment is randomly initialized at the beginning of each episode, and confounders remain unchanged until the end of the episode. Note that, DOMINO treats all the negative examples equally and doesn't need the specific value of the confounders. RIA doesn't need to record if the two trajectories are collected in the same episode, since the relational intervention approach could optimize the mutual information without environment labels and even without the environment ID, which provides a promising direction of unsupervised dynamics generalization.
> > >
> > >
> > > The advantage of DOMINO is that it can alleviate the problem of underestimation of mutual information caused by insufficient interaction data, which seriously affects MI optimization. DOMINO aims to alleviate this problem and optimize the mutual information effectively using as less data as possible. Here, we explain in detail:
> > >
> > > 1) When the number of confounders is increasing, the demand for data to let the $I_{NCE}(c;\mathcal{T})$ be a tight lower bound increase exponentially.  By decomposing the whole $I_{NCE}(c;\mathcal{T})$  into $\sum_{i=1}^{N} I_{NCE}(c_{i};T)$, and optimize each $I_{NCE}(c_{i};T)$ separately, while regularizing the mutual information $I(c_{i};c_{j})$ between each disentangled context vectors .The demand of the  data amount $K$ can be reduced from $e^{I(c;\mathcal{T})}$ to $e^{\frac{1}{N}I(c;\mathcal{T})}$.
> > >
> > > 2) The proposed decomposed MI optimization can benefit many contrastive learning-based context learning methods based on the InfoNCE. We believe that DOMINO can also benefit RIA to further improve the sample efficiency with multiple confounders.
> > >
> > >
> > > We believe that DOMINO and RIA are not in competition, on the contrary, their effective combination will become a stronger baseline, for example, the decomposed MI optimization can be expanded into the relational intervention approach proposed in RIA.
> > >
> > > **Q2: RIA only uses one prediction head as their codes show, so it does not add the intervention module to TMCL**
> > >
> > > ANS:
> > >
> > > Thanks for your suggestion. We provide the performance comparison between RIA and DOMINO without adaptive planning accordingly(see Figure 4, in section 5.2, third revised paper).
> > >
> > > **Q3: Can the authors run additional tests to verify that the suggested approach is reliable when is greater than the actual number of confounders?**
> > >
> > > ANS：
> > >
> > > Thanks for your constructive suggestion. We run the additional test to verify that the suggested approach is reliable when $N$ is greater than the actual number of confounders (see Figure 10, Appendix E.3, third revised paper). The results show that when $N$ is greater than the actual number of confounders, the performance is still better than T-MCL and is comparable with the other ablation version of DOMINO. Although the extra context may encode the information irrelevant to the confounder, the policy only needs to learn to pull out the useful ones from multiple context vectors and discard the useless ones, which can be learned by minimizing the prediction loss.

---

> ### Author Response · Authors · 2022-08-02
> **Response to Reviewer RkbU (Part 1/2)**
>
> Thank you very much for recognizing our idea, writing, and presentation. We sincerely thank you for the valuable suggestions. Here, we address your concerns as follows:
>
> **Q1. About the adaptation process and the relationship between dynamics generalization and Meta-RL.**
>
> ANS:
>
> We mention the adaptation process in the preliminaries. We learn a context encoder to capture the environmental information and a context-conditional policy to generate actions in the training process. At test-time, the policy zero-shot adapts to the new MDP under the unseen confounders setting $u_{test}$ conditioned on the inferred context. During the adaptation process, the context encoder maps the sequence of state action pairs in new dynamics into context vectors, and the policy generates actions condition on the learned context. Since the proposed decomposed mutual information optimization framework can be used as a plug-and-play module to combine with the conventional meta-reinforcement learning methods, and the key of this paper is to learn the better context encoder, we have introduced this part relatively little. With your kind suggestion, we have added a description of the adaption process related to it in the revised version.
>
> The generalization objectives of meta-reinforcement learning include several aspects such as generalization of tasks, generalization of robot dynamics, generalization of environments, etc. Dynamics generalization of reinforcement learning belongs to a branch of meta-reinforcement learning. For example,  in PEARL, which is a typical meta-reinforcement learning algorithm, the experiments on Walker-2D-Params domain are to test the performance of dynamics generalization.
>
> By conditioning on an effective context, Meta-RL policies can easily generalize to new tasks within a few adaptation steps.
> Context-based Meta-RL methods like PEARL (Titled: Efficient Off-Policy Meta-RL via Probabilistic Context Variables) and CCM (Titled: Towards Effective Context for Meta-Reinforcement Learning) then train a policy conditioned on the latent context to improve generalization. Both TMCL and CADM can be categorized as context-based meta-reinforcement learning methods.
>
> **Q2: About the novelty and the relationship with previous works**
> ANS:  The core innovation of this paper is to address the challenges for accurate estimation of MI posed by the combination of multiple confounders.
>
> ANS:
>
> Maximizing MI  has been verified by several earlier works, such as [1,2,3], to have an improvement in the single confounder setting.  CCM [1] adds mutual information optimization and designs an additional exploration policy to collect more elective data. FOCAL [2] introduces contrastive learning method to offline meta-RL. RIA [3] introduces a relational intervention approach to TMCL which also attempts to maximize the MI between context vector and historical trajectories.
>
> However, when multiple confounders act together, it is difficult to learn an accurate entangled context variable to cover all the information of multiple confounders. Our theoretical analysis also shows that as the number of confounders increases, InfoNCE may become a very loose lower bound, which poses a challenge for the optimization of MI. Therefore, this paper focuses on addressing such a challenge by decomposed MI optimization.
>
> Experimental Comparison:
>
> DOMINO has designed two parts of experiments combined with the model-based approach and with the model-free approach.
>
> In the experiments combined with the model-based method, both DOMINO and paper [3] are implemented based on TMCL.
> In the experiments combined with the model-free approach, both DOMINO and paper [1] are implemented based on PEARL.
>
> In this paper, we use TMCL and PEARL as the main baseline and add an ablated version that optimizes only one mutual information between the entangled context and the trajectory for comparison, aiming to highlight the effect of the proposed decomposed MI optimization method under the multi-confounded setting.
>
> With your kind suggestion, we add a comparison experiment to compare DOMINO and RIA. As shown in Figure 8 (**see Appendix E.2 in page21, revised version paper**), DOMINO achieves better generalization performance than RIA and TMCL, especially in complex environments like  Halfcheetah-$m$-$d$ and Slim-humanoid-$m$-$d$.
>
> [1] Haotian Fu, Hongyao Tang, Jianye Hao, Chen Chen, Xidong Feng, Dong Li, and Wulong Liu. Towards effective context for meta-reinforcement learning: an approach based on contrastive learning.
>
> [2] Li, L., Huang, Y., Chen, M., Luo, S., Luo, D., & Huang, J. (2021). Provably Improved Context-Based Offline Meta-RL with Attention and Contrastive Learning. arXiv preprint arXiv:2102.10774.
>
> [3] Guo J, Gong M, Tao D. A Relational Intervention Approach for Unsupervised Dynamics Generalization in Model-Based Reinforcement Learning[C]//International Conference on Learning Representations. 2022.

---

> ### Author Response · Authors · 2022-08-06
> **Sincerely looking forward to more discussion with you**
>
> Dear reviewer RkbU:
>
> We thank you for the precious review time and valuable comments. We have provided corresponding responses and results, which we believe have covered your concerns. We sincerely hope to further discuss with you whether or not your concerns have been addressed. Please let us know if you still have any unclear parts of our work. We are always ready to address your further concerns.
>
> Best,
>
> The authors.

---

### Official Review · Reviewer_1VJF · 2022-07-07

**Rating:** 6
**Confidence:** 4
**Soundness:** 3 good
**Presentation:** 3 good
**Contribution:** 3 good

**Summary:**

This paper tackles the problem of generalization in MDPs where the dynamics changes are assumed to be caused by multiple independent factors, denoted as context. The proposed framework (DOMINO) learns a context encoder that maps trajectories to a latent context via decomposed mutual information using noise-contrastive estimation (InfoNCE). The authors combine DOMINO with model-free and model-based RL algorithms, and perform experiments in classic environments, as well as in the Mujoco benchmark, in settings where multiple confounders change simultaneously. Additionally, qualitative visualizations of the latent context vectors are presented using t-SNE.

**Questions:**

Furthermore, I have the following questions and constructive criticisms:

- “Our goal is to learn a generalized context encoder, which is learned in training process by maximizing the expected rewards R_train in seen environments, and zero-shot generalized to unseen environments for a high expected rewards R_test”
This sentence is hard to follow. The objective is also not very clearly defined in Eq. 1. For instance, how is the context $c$ generated? How do the confounders $u$ affect the expectation?

- In Section 4.2, it is not clear the difference between the two context encoders, $g_\phi$ and $h_w$? Importantly, why is $h_w$ necessary? Notice that $h_w$ does not appear in the pseudo-code of the algorithms in the appendix.

- It is not clear what are the parameters being optimized through the loss functions in Eq. 5, 8, and 9. I suggest specifying which parameters ($\psi$, $\phi$, etc.) are involved in the gradients of each loss function.

- “The critic function $\psi(·, ·)$ measures the similarity between inputs by dot product.” Why was dot product chosen as the similarity measure? Can you elaborate on this decision?

- “The results illustrate that DOMINO learns the policy more efficiently and stably than the baselines.”
It is not possible to infer that DOMINO learns more “stably” than the baselines. In fact, sometimes it shows more variance (see Fig. 3 - Halfcheetah-m-d) than the baselines.

- Why does MINO perform better than T-MCL? It seems the improvements of MINO (in comparison to T-MCL) are more important than the improvements of DOMINO (in comparison to MINO).

- In Figure 6, it is not clear what each point represents. Given a trajectory, the context encoder outputs N different context vectors. In this figure, did you concatenate all vectors before applying t-SNE?

- The key characteristic of the proposed method is the fact that it should learn disentangled and independent latent contexts. However, the visualizations using t-SNE were not able to show that. An important result would be to show that indeed each latent context captures one of the confounders of the environment. This could be shown by varying only one of the confounders and observing whether only one of the latent contexts changes accordingly.

- The authors state in the checklist that they have included the code, data, and instructions needed to reproduce the main experimental results in Appendix D. However, only pseudocode of the algorithms and code for the environment were made available. The actual code of the proposed methods is not available.

- In the paper’s abstract, it is said that the open-sourced code and videos are released on their anonymous homepage. However, there are no videos on this page, and only code for the environments is available, not the algorithms/training code.


**Limitations:**

The paper could benefit from a discussion regarding the assumption of independent confounders. For instance, how difficult it would be to adapt the algorithm to the case where we have co-related confounders?

**Strengths And Weaknesses:**

Strengths:
- The idea of capturing the different confounders that may affect the dynamics of the MDP into different latent contexts is novel and interesting.
- The experimental results show that the proposed method can, in general, achieve better performance than the state-of-the-art.

Weaknesses:
- The paper needs improvement regarding the clarity of the mathematical definitions, such as the objective functions.
- It is not clear whether the improvements are because of the decomposed mutual information framework, or because of other algorithmic improvements (see below).

---

> ### Author Response · Authors · 2022-08-02
> **Response to Reviewer 1VJF (Part2/2)**
>
> **Q5. About the stability of DOMINO learning process.**
>
> ANS:
>
> Sorry for the confusion. We aim to claim that DOMINO is less likely to deteriorate after learning a good policy. There is less oscillation in the DOMINO training process compared with TMCL, and the variance shown in the figure is caused by the difference in performance under different random seeds, and the learning process under each seed shows less oscillation compared to TMCL. For example, as shown in Figure 3 and Figure 4, in the slim-humanoid-m-d, even with 1.6*1e5 steps, TMCL is also likely to perform poorly, while DOMINO  always performs well. Although it has more variance between different seeds in Halfcheetah-m-d, it performs better than the baseline, and the return curve is always above the baseline.
>
>
> **Q6. Why does MINO perform better than T-MCL?**
>
> ANS:
>
> T-MCL learns the context encoder only under the supervision of transition prediction, while MINO learns the context encoder via both the transition prediction and mutual information optimization and could help the context encoder to extract better environmental information. Previous works like CCM and TCL also prove that adding mutual information optimization could help context learning under a single confounder setting. This paper aims to reveal the advantages of decomposed mutual information optimization in the multi-confounder setting compared to the approach that only optimizes entangled context with historical trajectory mutual information. Therefore, it is an important ablation version.
> In terms of the experimental results, MINO may perform worse than TMCL in the environments like Cartpole-f-l, while DOMINO performs well. Furthermore, DOMINO shows more advantages than MINO in the experiments combined with model-free methods.
>
> **Q7. In Figure 6, did you concatenate all vectors before applying t-SNE?**
>
> ANS:
>
> Yes, we concatenate all context vectors together and visualize them in the 2D space by t-SNE.
>
>
> **Q8. The effect of t-SNE**
>
> ANS:
>
> The visualization aims to show whether the contexts can be separated from each other under multi-confounder setting, which is one of the key factors of the performance of the conditional policy.
>
> **Q9. Please show that  each latent context captures one of the confounders of the environment**
>
> ANS:
>
> Thanks for your good suggestion. We add an additional experiment to vary only one of the confounders and observe the changes of $N$ disentangled vectors accordingly.
> In this experiment, we set up two different confounders:  mass $m$ and damping $d$. Under the DOMINO framework, the context encoder inferred two disentangled context vectors: context 0 and context 1.
> As shown in Figure 10 and Figure 11 (**see Appendix F.1, page 22, the revised version paper**), context 1 is more related to damping. When the confounders are set as the same mass but different damping, the visualization result of context 1 under different settings are separated clearly from each other, while under the same damping but different mass settings, the visualization result of context 1 is much more blurred from each other. Similarly, context 0 is more related to mass. When the confounders are set to the same damping but different mass, the visualization result of context 0 under different settings is separated clearly from each other, while under the same mass but different damping settings, the visualization result of context 0 is less different from each other.
>
> **Q10. About the training code**
>
> ANS:
>
> The whole training code is available at the following anonymous link: https://anonymous.4open.science/r/DOMINO_NIPS-CEC1/

---

> > ### Comment · Reviewer_1VJF · 2022-08-05
> > **Response to Authors**
> >
> > I thank the authors for carefully addressing my questions and concerns.
> >
> > Q1) Minor: Although I understand the textual problem formulation, I believe the mathematical definition can be better formalized. Eq.1 is only defining the return, but not the optimization problem. For instance, it would be clearer for readers if there was a $\max_\pi$ in Eq.1.
> >
> > Q9) I appreciate the author's effort in providing this visualization. It better elucidates the method's capabilities of learning disentangled contexts
> >
> > I also share reviewer oNNM concern (in their latest reply) regarding the tight bound for $I_{NCE}$. Would you please clarify whether the number of data necessary to make $I_{NCE}$ a tight bound is not impractical?

---

> > > ### Author Response · Authors · 2022-08-06
> > > **Response to your further concerns**
> > >
> > > Thanks for your kind reply. We carefully address your concerns as follow:
> > >
> > > **Q1:
> > > Eq.1 is only defining the return, but not the optimization problem. For instance, it would be clearer for readers if there was a $\max_{\pi}$**
> > >
> > > ANS:
> > >
> > > Thanks for your kind suggestion. We define the optimization problem as follow:
> > > We aim to learn the optimal policy condition on the context $c$ encoded from the current sequence of state action pairs $[s_{\tau}, a_{\tau}, s_{\tau+1}]_{\tau=t-H}^{t}$.
> > >
> > > \begin{equation}
> > > \max_{\pi} E_{\tilde{u} \sim p(\tilde{u} \_{\\#})}[\sum_{t=0}^{\infty} \gamma^{t} r\left(s_{t}, \mathbf{a}_{t}\right)]
> > > \end{equation}
> > >
> > > where $a_{t} \sim \pi(s_{t}, c), \\#=[\text { ''train"or ''test" }]$ and $\tilde{u} =[u_{0},u_{1},\ldots,u_{N}]$ represents the multiple confounders.
> > >
> > > We also revised the paper accordingly with your kind suggestion(**see Equation 1, page 3, second revised version paper**).
> > >
> > > $$$$
> > >
> > > **Q2: Would you please clarify whether the number of data necessary to make  a tight bound is not impractical?**
> > >
> > > ANS:
> > >
> > > For a disentangled context, the demand of data amount $K$ needs to satisfy $K \geq e^{\frac{1}{N} \sum_{i=1}^{N} I(u_{i},\mathcal{T})}$.
> > >
> > > For a entangled context, the demand of data amount $K$ needs to satisfy $K \geq e^{\sum_{i=1}^{N} I(u_{i},\mathcal{T})}$.
> > >
> > > It means that the demand data in the methods using entangled context is far greater than DOMINO. As the number of confounders increases, this gap will widen exponentially.
> > >
> > > In addition, since RL has high requirements for sample efficiency, we want to use as little data as possible to learn the policy. Therefore, DOMINO has an obvious advantage over methods that use entangled contexts.
> > >
> > > We again appreciate your discussion and valuable comments.
> > >
> > > We submitted the second revision of our paper.
> > > The first revision: blue The second revision: red
> > >
> > > We are always ready to address your further concerns and revise the paper if you suggest anything that can improve the quality of our paper.

---

> > > > ### Author Response · Authors · 2022-08-09
> > > > **Looking forward to your reply**
> > > >
> > > > Dear Reviewer 1VJF:
> > > >
> > > > We sincerely appreciate your constructive suggestions to improve the quality of our articles. We hope we resolve all of your concerns and we wish you could reconsider your score.
> > > >
> > > > Best,
> > > >
> > > > The authors.

---

> > > > > ### Comment · Reviewer_1VJF · 2022-08-09
> > > > > **Response**
> > > > >
> > > > > I thank the authors for further clarifying my questions and concerns, and for the effort to improve the paper according to the suggestions made by me the other reviewers.
> > > > > Hence, I increased my score as the paper were significantly improved during the rebuttal period.

---

> ### Author Response · Authors · 2022-08-02
> **Response to Reviewer 1VJF (Part1/2)**
>
> Thank you for your recognition of the novelty of our method and the experimental results. Since your concern is mainly related to the clarity of the mathematical definitions, we provide a more detailed explanation of them and polished our paper with your kind suggestion. Here, we address your concerns as follows:
>
> **Q1. Explanation of the objectives and how confounder affects the  $R_{train}$ and  $R_{test}$.**
>
> ANS:
>
> We want to learn a policy condition on the context, which is encoded from the sequences of current state action pairs $[s_{\tau},a_{\tau},s_{\tau+1}]_{\tau=t-H}^{t}$ in several training scenarios and enable it to perform well in test scenarios never seen before.
>
> Specifically, the agent infers the context $c$ that characterizes the current environment through a context encoder $g_{\phi}$ from the historical trajectories obtained from interactions in the current environment and then generates actions conditioned on the context $c$.
>
> The characteristics of the training scenario are determined by confounder $u$. The confounder $u$ affects the dynamics of the robot and thus the distribution of the state transfer $(s,a,s')$, and the reward as a function of $(s,a,s')$, thus will affect the $R_{train}$ and $R_{test}$.
> In the test scenario, the settings of the confounder $u$ are different from those in the training scenario, and we hope that the learned context encoder has good generalization performance so that the policy conditioned on the learned context can also achieve high returns in the test environment.
>
> **Q2. the difference between the two context encoders.**
>
> ANS:
>
> The context encoder $g_{\phi}$ aims to encode the sequence of state action pairs in the current episode into the $N$ context vectors $c_{1},c_{2},...,c_{N}$, while the function $h_{w}$ aims to map the historical trajectory into the same dimension of $c_{i}$. The key difference is the output dimension of $g_{\phi}$ is $N$ times than $h_{w}$.
>
> **Q3. the parameters being optimized through the loss functions in Eq. 5, 8, and 9.**
>
> ANS:
>
> Sorry for the confusion. The parameters being optimized in Equation5, is the $\varphi$ of context encoder $g_{\varphi}$ and $w$ in function $h_{w}$ which maps the trajectory into the same dimension of $c_{i}$.
>
> The parameters being optimized in Equation8, is the $\varphi$ of context encoder $g_{\varphi}$ and the $\phi$ in the prediction model $f_{\phi}$.
>
> The Equation 9 is a combination of Equation 5 ($L_{NCE}$) and Equation 8 ($L_{Pre}$), and the parameters being optimized  are $\varphi$, $w$ and $\phi$.
>
>
> **Q4.Why was the dot product chosen as the similarity measure ?**
>
> ANS:
>
> The critic function is calculated by the dot product of the input vectors with normalization. Thus the output is the cosine similarity of two input vectors.

---

### Official Review · Reviewer_rYrP · 2022-07-11

**Rating:** 7
**Confidence:** 4
**Soundness:** 2 fair
**Presentation:** 3 good
**Contribution:** 2 fair

**Summary:**

This paper studies a contextual reinforcement learning (RL) setting where the environment dynamics are parameterized by independent factors, which the authors refer to as “confounders.” In each episode, the underlying factors can vary. They present a method for contextual meta-reinforcement learning (RL) called DOMINO, which learns to encode the RL agent’s current trajectory into a set of independent context vectors. These independent context vectors can then be used as inputs to the transition model in model-based RL (MBRL) and as an input to the policy in model-free RL, thereby providing the agent with an inferred context for the underlying environment factors in any given episode. Importantly, their method assumes the underlying environment factors are similarly independent. The main contributions of the paper are the method, DOMINO, for learning independent context vectors from the trajectory and their analysis and experimental results demonstrating the favorable properties of this method (including improved empirical performance against baselines learning entangled context vectors), when the underlying independence assumptions are valid.

**Questions:**

- Could the authors elaborate on their choice of T-MCL as the sole baseline in their MBRL experiments? Further details on this choice would benefit the clarity of the experiments section.
- Further, since the goal of the method is to perform efficient meta-test adaptation via context vectors, why was PEARL chosen over Varibad, which has been shown to provide much more efficient within-episode adaptation compared to PEARL?
- It seems that the number of context vectors N must be set to the number of environment factors. Is this understanding correct?
- This paper assumes the environment confounders impact the transition function, but not the reward function. How do the authors view the role of reward generalization in their work? Can DOMINO be expected to work in settings where the environment confounders also impact the reward function?

**Limitations:**

The core assumption of this work also acts as its primary limitation: The environment factors of variation are assumed to be independent, and their number known a priori. The authors should make an effort to emphasize this limitation and to what extent they believe such an assumption of independence may be applicable in practice.

**Strengths And Weaknesses:**

Strengths

- The paper provides a simple method for improving context-aware meta RL in an environment with multiple independent factors of variation that impact the transition dynamics. The method itself is clearly described. This seems to be the first method to directly exploit an explicit assumption of independence among the underlying environment factors of variation.
- The method performs well against sensible baselines. Importantly the method performs well against an ablation that does not learn disentangled context vectors.

Weaknesses

- The reported results in the Table 1 and 2 have high overlap between the authors’ DOMINO and MINO methods and the baselines. The signficance of these results could be made clearer by reporting the results of a Welch t-test between the proposed method and the baselines.
- Similarly, the performance comparison plot in Figure 1b should have error bars. It should also state what method of averaging was used for the plotted values
- The paper can benefit from a full pass to improve the clarity of the writing. There are numerous missing details about basic figures, such as what measure of uncertainty is represented by the error bars for each plot and table. There are also several ambiguous phrasings and sentences with confusing wording. For example
- A key aspect of this paper is the analysis of InfoNCE as a “loose bound” of the mutual information. However, the authors never define whether this bound is an upper or lower bound. While this detail can be inferred from context, I think it is important to make this point clearer to the reader. Relatedly, the definition of “MI underestimation” in L45 is unclear.
- Given that the independence assumption is core to this work, it is unclear how significant this setting will be in practice and for future work.
- Moreover, it seems important for the experiments to assess how valid such an independence assumption is in practice, and crucially, what is the price in performance one might expect to pay for making this assumption. An experiment assessing the performance of DOMINO and MINO on a more complex environment whose underlying factors of variation are not mutually independent would improve this paper by providing a more complete picture of the effectiveness of this method.
- There seems to be an underlying assumption that the N independent context vectors aim to encode information about the underlying factors of variation in the environment. However, this connection is actually never explicitly made in the writing, making the jump from discussing MI in terms of environment factors to context vectors (4.1 to 4.2) unclear.
- It seems that DOMINO requires setting the number of context vectors N equal to the number of environment factors of variation. In general, we may not know this value exactly. Adding a sensitivity analysis to how dependent the performance is on setting N to this exact value would provide important information on how applicable this method is in practice.

Minor comments:

- L22: “mythologies” should be “morphologies”.
- L47-48: “First the context encoder embeds the past state-action pairs into disentangled context vectors” is an inaccurate description, as it must first be optimized to do so (as next described in L48-49).
- This paper could consider citing related work in unsupervised environment design [1,2,3,4] and more generally, RL work in procedurally-generated environments [5,6]. These works are deeply related as they effectively perform meta-RL over a space of environment variations with an implicitly learned context. Ignoring this line of work seems like a significant oversight.

References

[1] Dennis et al, 2020. Emergent Complexity and Zero-shot Transfer via Unsupervised Environment Design.

[2] Jiang et al, 2021. Prioritized Level Replay.

[3] Jiang et al, 2021. Replay-Guided Adversarial Environment Design.

[4] Parker-Holder et al, 2022. Evolving Curricula with Regret-Based Environment Design.

[5] Raileanu et al, 2021. Decoupling Value and Policy for Generalization in Reinforcement Learning.

[6] Cobbe et al, 2019. Leveraging Procedural Generation to Benchmark Reinforcement Learning.

---

> ### Author Response · Authors · 2022-08-02
> **Response to Reviewer rYrP [questions listed in "weaknesses" section](Part2/2)**
>
> **Q6: what is the price in performance one might expect to pay for making this assumption?**
>
> ANS:
>
> If the confounder independence assumption is no longer satisfied, splitting the mutual information into sums of several subitems to be estimated separately will lead to an overestimation of the mutual information. And fortunately, the penalty term in $L_{NCE}$ loss function for $I(c_{i},c_{j})$ can suppress this phenomenon to some extent. Therefore, the performance does not deteriorate in more complex environments, where the assumption is not satisfied, compared to baselines such as TMCL.
>
> **Q7:Adding a sensitivity analysis to how dependent the performance is on setting N.**
>
> ANS:
>
> Thanks for your constructive suggestion. We added the sensitivity analysis on the hyper-parameter $N$, and have added it to the revised version.
>
> We compare the performance of DOMINO with different hyper-parameter $N$, which is equal or not equal to the number of confounders in the environment. In this experiment, the confounder is the damping, mass, and a crippled leg (number of confounders is 3), and we compare the performance of DOMINO with different hyper-parameter $N={1,2,3}$.
> As shown in Figure 9 (**see Appendix E.3, page21, revised version paper**), even though the hyper-parameter $N$ is not equal to the ground truth value of the confounder number, DOMINO also benefits the context learning. In practice, under a conservative setup of hyper-parameter $N$, DOMINO can also benefit the context learning compared to the baselines like TMCL.

---

> > ### Comment · Reviewer_rYrP · 2022-08-06
> > **Response to authors' rebuttal**
> >
> > Many thanks for the detailed clarifications on each of my questions. I especially appreciate the usage of colored diffs in the updated manuscript. I am thus increasing my rating to fully support "Accept."
> >
> > Given the extra page for the CRC, think it would be valuable for you to include these points from your response in the final version of the paper:
> > - How you envision VariBAD can be combined with DOMINO to further improve context adaptation.
> > - How you envision DOMINO can be extended to reward generalization.
> > - How DOMINO can be extended to support the case of dependent factors of variation in the environment.
> > - Include a reference to your RIA comparisons in the Appendix, or even incorporate them into the main results.

---

> > > ### Author Response · Authors · 2022-08-09
> > > **Sincerely thank you for your recognition and support of our paper**
> > >
> > > We sincerely thank you for your recognition and support of our paper. We have added content to the revised paper based on your suggestions. The constructive suggestions you gave during the rebuttal session are greatly helpful in improving the quality of our paper, thanks to your hard work and in-depth discussions with us.

---

> ### Author Response · Authors · 2022-08-02
> **Response to Reviewer rYrP [questions listed in "weaknesses" section](Part1/2)**
>
> **Q1: Reporting the results of a Welch t-test between the proposed method and the baselines**
>
> ANS:
>
> Thanks for your suggestion, we have added the Welch t-test results between the the proposed method and baselines in the revised version paper. The p-value  results of the Welch t-test between the proposed method and the baselines  are shown as follows:
>
> |       | cartpole | pendulum   | ant      | halfcheetah | slimhumanoid | hopper   |
> |-------|----------|------------|----------|-------------|--------------|----------|
> | TMCL  | 1.27e-11 | 7.1776E-10 | 0.000756 | 0.010593    | 0.134583     | 0.019499 |
> | MINO  | 3.87E-07 | 0.000619   | 0.006396 | 0.006166    | 0.108993     | 0.000698 |
> | PEARL | 0.000824 | 0.003779   | 0.053568 | 0.041328    | 5.4165E-05   | 0.011843 |
>
>
> Thus, with significance level $\alpha=0.15$, the improvement of DOMINO compared to the baselines are significant ($p<0.15$).
>
> **Q2: the performance comparison plot in Figure 1b should have error bars. It should also state what method of averaging was used for the plotted values**
>
> ANS
>
> Thanks for your kind suggestion. We have revised Figure 1b with error bars(see page2, revised version paper). The method of averaging is that we calculated the average return of 20 random tests.
>
> **Q3:what measure of uncertainty is represented by the error bars for each plot and table?**
>
> ANS:
>
> The error bars for each plot and table show the confidence interval of the average return with 20 random tests over different random seeds. Since reinforcement algorithms are usually sensitive to the seed, most of the papers show the average return with confidence intervals over more than 5 random seeds. In the shown figures, the line represents the average value of different seeds, and the shade represents the confidence interval, which is calculated by a typical data visualization package Seaborn.lineplot.
>
> **Q4:whether this bound is an upper or lower bound and what is the definition of “MI underestimation”**
>
> ANS:
>
> The InfoNCE bound  is a lower bound of the mutual information, and the MI underestimation in our original paper is defined as follows:
> When the InfoNCE bound is much smaller than the mutual information, which is also called underestimation, then the InfoNCE is a loose lower bound, and in this situation, the MI is hard to be optimized by maximizing the InfoNCE bound according to underestimation.
>
>
> **Q5:Given that the independence assumption is core to this work, it is unclear how significant this setting will be in practice and for future work.**
>
> ANS:
>
> In practice, the dynamics generalization problem faced by the robot is mainly caused by the mismatch between various dynamics parameters in the simulation environment and the real world, such as mass, damping coefficient, length, size, and stiffness of the mechanical structure. These parameters themselves are usually independent, and when many of them are different in both real-world and simulation environments, the robot dynamics will exhibit more complex variations, posing a great challenge to the generalization of control strategies.
>
> Therefore, based on this assumption, inferring information about each individual confounder from state transition sequences is very important for improving the performance of the robot. This process is similar to the process in which a human calibrates each kinetic parameter respectively, except that DOMINO lets the relevant information be encoded from the trajectory automatically using a context-based approach.
>
> For future works, we can explore how to extract the information that is most useful for state transfer from each of the confounders separately when they do have some correlation with each other. One possible option is to adjust the penalty factor for mutual information between the context vectors in DOMINO, which can be set to be dynamically adjustable.

---

> ### Author Response · Authors · 2022-08-02
> **Response to Reviewer rYrP [questions listed in "questions" section]**
>
> We appreciate your careful reading of our and your very constructive suggestions. Here, we address your concerns as follows.
>
> **Q1:Why choose T-MCL as the sole baseline in their MBRL experiments**
>
> ANS:
>
> We choose TMCL as the main baseline because: 1) TMCL is the state-of-the-art method in model-based meta-RL. 2) TMCL and its base version CADM are the first to provide a rich meta-RL benchmark affected by confounders (containing both discrete and continuous confounders). In contrast, all other methods give results in only a small number of environments. 3) RIA is a recent advanced algorithm, which is concurrent work of DOMINO and is also implemented based on the TMCL and has not yet open source when we ran our comparison experiments. Since RIA has open-sourced the code, we have added the relevant comparison experiments with RIA.
>
>
> **Q2:why was PEARL chosen over Varibad?**
>
> ANS:
>
> The experiments combined with the model-free approach aim to verify the degree of improvement of decomposed MI optimization and entangled MI optimization under the multi-confounder setting, respectively, compared to relying only on the reward signal to learn the context, and thus to validate the advantages of DOMINO.
> PEARL is the most typical and widely used algorithm that relies only on the forward signal learning context and is also used as a key baseline in TMCL and CADM. Varibad introduces the VAE method and recurrent network to learn the context, which optimizes the context learning from different perspectives from DOMINO and TMCL methods and does not conflict with each other. Thanks to your constructive suggestions, we think the effective combination of DOMINO and Varibad will become a more powerful baseline for meta-RL.
>
> **Q3:It seems that the number of context vectors N must be set to the number of environmental factors. Is this understanding correct?**
>
> ANS:
>
> Yes, theoretically, when the number of context vectors N must be set to the number of environmental factors, the performance is the best. Here, we also provide the experiment about the DOMINO’s sensitivity testing to N. We find that DOMINO still performs better than TMCL even when N is not consistent with the true number of confounders. As long as N is set to a respectively conservative value that does not exceed the total number of potential confounders in the environment, DOMINO can still benefit the context learning.
>
> **Q4:How do the authors view the role of reward generalization in their work? Can DOMINO be expected to work in settings where the environment confounders also impact the reward function?**
>
> ANS:
>
> Really good question! This is a very good research direction!
> The reward generalization can be categorized as a kind of task generalization.
> The parameter of the reward function, for example, the target speed of the robot, can also be considered as a confounder that influences the reward transition.
> To address this problem under the DOMINO framework,  we provide the following solution.
> The context encoder maps the current sequence of state-action-reward pairs $[s\_{\tau},a\_{\tau},r\_{\tau}]^{t}\_{t-H}$ i into disentangled contexts, which contains the information of the physical confounders like mass and damping and the reward confounder. The historical trajectory also should consider the reward part, i.e., $s_t,a_t,r_t$,$s\_{t+1}$. Then the proposed decomposed mutual information optimization method can also be used in this situation to extract effective context. Moreover, the prediction loss should also add the reward prediction term. Thus, with the above design, DOMINO can address the reward generalization and dynamics generalization simultaneously.

---

### Official Review · Reviewer_oNNM · 2022-07-11

**Rating:** 6
**Confidence:** 4
**Soundness:** 3 good
**Presentation:** 3 good
**Contribution:** 2 fair

**Summary:**

This paper addresses the problem of learning generalizable context in RL. In particular, it suggests learning disentangled context representation of each confounding in the environment using the proposed model, DOMINO, which optimizes decomposed MI objectives. It adopts the contrastive learning method when learning the disentangled context representation, regarding trajectories sampled from the setting of the same confounding as positive pair and of different confounding as negative pair. The authors also provide a theoretical basis for how optimizing their decomposed MI objective can make $I_{NCE}$ a tighter lower bound by alleviating the underestimation of MI. By learning policy conditioning on the learned context vector, DOMINO can achieve higher generalization performance compared to both model-based and model-free baselines.

**Questions:**

- In Appendix A,
  - How does the first equility ($q(y | x, y_{2:K}) = p(y) K w_y$) hold?
  - What is $w_y$?
  - What does $E$ mean in $I_{NCE}(x;y | E, K)$?
- In Theorem 1,
  - Even if the number of confounders increases, the true mutual information $I(c; \mathcal T)$ does not. Therefore it shows inconsistency to regard $I_{NCE}(c; \mathcal T)$ and $I_{NCE}(c_i; \mathcal T)$ as having the same upper bound.
  - The following inequality also holds in the same setting. $\sum I_{NCE}(c_i;\mathcal T | K) = I_{NCE}(c;\mathcal T| K) \le log K$
- In Figure 6,
  - DOMINO encodes the context vector of each confounder mass and damping. However, Figure 6 does not show how effectively each confounder was encoded.
  - For example, Setup0 and Setup4 have the same damping condition and Setup0 and Setup3 have different mass and damping conditions. Then shouldn't Setup0 and Setup4 be located closer than Setup0 and Setup3?

**Limitations:**

Yes, the authors adequately addressed the limitations and potential negative social impact of their work.

**Strengths And Weaknesses:**

Strengths:

The paper is well written and clear to understand. Using contrastive loss when learning disentangled representation of each confounding is novel and intuitive. And it is intriguing to get an idea of sampling negative pairs from different episodes. The experiments are comprehensive and the results are impressive.


Weaknesses:

However, the proof of Lemma 1 and Theorem 1 lacks mathematical rigor. Also, there is some missing specific information about notations in the proof, thereby undermining the clarity and soundness of the paper (e.g., $w_y$ and $E$). Visualization of the learned context embeddings does not show how effectively each confounding is encoded.

---

> ### Author Response · Authors · 2022-08-02
> **Response to Reviewer oNNM (Part 3/3)**
>
>
> **Q3:Figure 6 does not show how effectively each confounder was encoded, shouldn't Setup0 and Setup4 be located closer than Setup0 and Setup3?**
>
> ANS:
>
> For context-based method methods, the core factor that influences the generalization performance is whether the context can separate the different settings as much as possible. Figure 6 shows that the context learned by DOMINO can be more easily distinguished than TMCL. Previous works with context visualization also used similar evaluation criteria as we did, i.e., the latent contexts from the same tasks are close in the embedding space while maintaining clear boundaries between the different tasks(see Section 5.3 in CCM paper).
>
>
> Since we consider all trajectories different from the current setting as negative cases when computing InfoNCE, and treat them equally when computing $L_{NCE}$, our current algorithm can only guarantee that the contexts under different settings can be separated obviously, and do not guarantee that similar settings will be encoded more similarly.
>
> The requirement kindly suggested by you that similar settings should be encoded in similar contexts and meanwhile distinguished well from each other is indeed a promising research direction, which can be left as future works.

---

> > ### Comment · Reviewer_oNNM · 2022-08-05
> > **Further questions**
> >
> > Thank you for the author's detailed explanation.
> > However, there are still some points to be clarified.
> >
> > 1. In the response Part 1/3, which theorem or lemma or equation did you bring from [2] and [3]? It seems inappropriate and unclear to cite the paper itself as above in developing the equations.
> > 2. I got the point of your claim in Part 2/3, however, don't we need N times of $K \ge e^{{1 \over N} \sum I(u_i; \mathcal T)}$ data to make every $I_{NCE}(c_i;\mathcal T)$ to be a tight bound?
> > 3. In the response Part 3/3, CCM paper refers to "Towards Effective Context for Meta-Reinforcement Learning: an Approach based on Contrastive Learning", right? Unlike CCM, DOMINO proposes a new framework for disentangling context representation of each confounding. Then, shouldn't the visualization be able to identify such disentanglement? In fact, the visualization results of T-CML, in which the expressions of Setup0 and Setup4 are intertwined, seem to be more reasonable.

---

> > > ### Author Response · Authors · 2022-08-06
> > > **Response to your further concerns (Part 2/2)**
> > >
> > > **Q3: shouldn't the visualization be able to identify such disentanglement. In fact, the visualization results of T-CML, in which the expressions of Setup0 and Setup4 are intertwined, seem to be more reasonable.**
> > >
> > > ANS:
> > >
> > > - The visualization in Figure6 is produced by concatenating the disentangled context vectors together with t-SNE method. This visualization aims to show whether the learned whole context, which contains all disentangled context vectors can be separated well under different settings.  The ability to separate the concatenated context vectors under different settings is an important prerequisite for the conditional policy to learn the optimal policy under each setting.
> > >
> > > - The phenomenon that the learned contexts in TMCL are intertwined in setup0 and setup4 is detrimental to policy learning, since the MDPs corresponding to setup0 and setup4 are different, if the contexts are intertwined, the policy that is conditioned on the context will learn a suboptimal solution to compromise between setup 0 and setup4. On the contrary, the whole context learned by DOMINO is clearly distinguished between setup 0 and setup 4, then the context-based policy can approximate the optimal solution under each of the two setups. Furthermore, note that setup 0 and setup4 are just similar in Hopper-m-d domain, the visualization in Cripple-Ant-m (Figure 6b), Setup0 (m = 1.15, leg = 3) and Setup4 (m = 1.0, leg = 2), which are quite different, while TMCL also inferred intertwined contexts.
> > >
> > >
> > > - Following your suggestion, we also think it is important to add a visual analysis for verifying whether the contexts is disentangled. Thanks for your constructive suggestion. Accordingly, we add an additional experiment which contains multiple confounders, and we only vary one of the confounders every time and observe the changes of disentangled vectors accordingly.
> > >     - In this experiment, we set up two different confounders:  mass $m$ and damping $d$. Under the DOMINO framework, the context encoder inferred two disentangled context vectors: context 0 and context 1.
> > >     - As shown in Figure 10 and Figure 11 (**see Appendix F.1, page 22, the revised version paper**), the context 1 is more related to damping. When the confounders are set as the same mass but different damping, the visualization results of context 1 under different settings are separated clearly from each other, while under the same damping but different mass settings, the visualization results of context 1 are much more blurred from each other. Similarly, context 0 is more related to mass. When the confounders are set to the same damping but different mass, the visualization results of context 0 under different settings are separated clearly from each other, while under the same mass but different damping settings, the visualization results of context 0 are less different from each other.
> > >
> > >
> > > We again appreciate your discussion and valuable comments.
> > >
> > > We submitted the second revision of our paper.
> > > The first revision: blue The second revision: red
> > >
> > > We are always ready to address your further concerns and revise the paper if you suggest anything that can improve the quality of our paper.

---

> > > > ### Author Response · Authors · 2022-08-09
> > > > **Thanks for your efforts and look forward to your reply.**
> > > >
> > > > We sincerely thank you for your efforts in reviewing our paper and your constructive suggestions again.
> > > >
> > > > We hope we have resolved all the concerns and showed the improved quality of the paper.
> > > >
> > > > And we deeply appreciate that if you could reconsider the score accordingly.
> > > >
> > > > We are always willing to address any of your further concerns.

---

> > > > > ### Comment · Reviewer_oNNM · 2022-08-09
> > > > > **Remaining concerns**
> > > > >
> > > > > Thank you for your detailed response to my questions.
> > > > > However, I still have remaining concerns on Theorem 1.
> > > > >
> > > > > To make each $I_{NCE}(c_i;\mathcal T)$ be a tight bound, we need $K \ge e^{I(u_i; \mathcal T)}$ amount of data.
> > > > > We are going to optimize $I_{NCE}(c_i;\mathcal T)$ by using separate negative and positive pairs for each context $i$.
> > > > > Then, don't we need $N$ times of $K \ge e^{I(u_i; \mathcal T)}$ amount of data?
> > > > >
> > > > > The authors clarified that $I_{NCE}(c;\mathcal T)$ and $I_{NCE}(c_i;\mathcal T)$ have different upper bound. And the authors are using the same upper bound $log K$ for both cases. I understood this $K$ as an arbitrary variable that represents the amount of data we need. So, the same $K$ does not mean the same amount of data, right? Then summing up $K$ in the proof of Theorem 1 does not make sense.

---

> > > > > > ### Author Response · Authors · 2022-08-09
> > > > > > **Response to your further concerns**
> > > > > >
> > > > > > Thanks for your reply. I consider that all your concerns of Theorem  1 come from the data we used to learn disentangled context vectors $c_{1},c_{2},…,c_{N}$.
> > > > > > We use the **same data** to infer disentangled context vectors $c_{1},c_{2},..,c_{N}$, since the state transitions in the trajectory are influenced by all the confounders.
> > > > > >
> > > > > > We carefully clarify as follows:
> > > > > >
> > > > > > - DOMINO infers a multiple disentangled context vectors $c_{1},c_{2},…,c_{N}$ from a single trajectory whose state transition is influenced by multiple confounders, and the concatenation of the disentangled context vectors $c=[c_{1},c_{2},…,c_{N}]$ represents the dynamics characteristic of the input trajectory. Therefore, we calculate each $I_{NCE}(c_{i};\mathcal{T})$ by using **same negative and positive trajectory pairs**.
> > > > > > Thus, we don’t need N times $K \geq e^{I(u_{i };\mathcal{T})}$ amount of data.
> > > > > >
> > > > > >
> > > > > > - DOMINO divides the mutual information problem into the summation of several smaller ones, which optimizes the mutual information between every disentangled context vectors and historical trajectory separately and regularize the mutual information between the disentangled context vectors. We decompose the calculation of $I_{NCE}(c;\mathcal{T})$ into the summation of $I_{NCE}(c_{i};\mathcal{T})$ with the **same data** which contains positive and negative pairs. Thus, we use same $\log K$ for both case.
> > > > > >
> > > > > > Thanks for your effort to discuss with us. Looking forward to your reply!

---

> > > > > > > ### Author Response · Authors · 2022-08-09
> > > > > > > **Looking forward to your reply!**
> > > > > > >
> > > > > > > We believe that your concerns about the theorem can be addressed by the above clarifications.
> > > > > > >
> > > > > > > We hope that we can resolve all of your concerns.
> > > > > > >
> > > > > > > We sincerely thank your effort to review our paper, and we would sincerely appreciate it if you could reconsider your score accordingly.

---

> > > > > > > > ### Comment · Reviewer_oNNM · 2022-08-09
> > > > > > > > **Reply to the authors**
> > > > > > > >
> > > > > > > > Thanks for the authors' detailed comment. I think that the last comment resolved my concerns about the clarity problem of theorem 1, so I increase my score from 3 to 6 accordingly.

---

> > > > > > > > > ### Author Response · Authors · 2022-08-09
> > > > > > > > > **Sincererly thank you for your recognition**
> > > > > > > > >
> > > > > > > > > We sincererly thank you for your constructive suggestions and effort. We appreciate your patience and in-depth discussions with us, and the quality of our paper has improved a lot with your help. Many thanks for your kind suggestions and warm help.

---

> > > ### Author Response · Authors · 2022-08-06
> > > **Response to your further concerns (Part 1/2)**
> > >
> > > We sincerely thank you for your kind reply. We carefully address your concerns as follow:
> > >
> > > **Q1: In the response Part 1/3, which theorem or lemma or equation did you bring from [2] and [3]? It seems inappropriate and unclear to cite the paper itself as above in developing the equations.**
> > >
> > >
> > > ANS:
> > >
> > > Thanks for your kind suggestion. We take [2] and [3] as reference to derive the proof of Lemma 1, we cite the equation we refer from [2] and [3] clearly as follow:
> > >
> > > According to the section 2.3 in [2], by setting the proposal distribution as the marginal distribution $\pi(y) \equiv p(y)$, the unnormalized density of $y$ given a specific set of samples $y_{2: K}=[y_{2}, \ldots, y_{K}]$ and $x$ is:
> > >
> > > \begin{equation}
> > > q\left(y \mid x, y_{2: K}\right)=p(y) \cdot \frac{K \cdot e^{\psi(x, y)}}{e^{\psi(x, y)}+\sum_{k=2}^{K} e^{\psi\left(x, y_{k}\right)}}=Kp(y)w_{y}
> > > \end{equation}
> > > where $K$ denotes the numbers of samples.
> > >
> > > According to the equation 3 of section 2 in [3], the expectation of $q(y \mid x, y_{2: K})$ with respect to resampling of the alternatives $y_{2: K}$ from $p(y)$ produces a normalized density:
> > > \begin{equation}
> > > \bar{q}(y \mid x)=E_{p\left(y_{2: K}\right)}\left[q\left(y \mid x, y_{2: K}\right)\right]
> > > \end{equation}
> > >
> > > We also revised the paper accordingly with your kind suggestion(**see Appendix A.1, page 14, second revised version paper**).
> > >
> > >
> > > $$$$
> > >
> > > **Q2: don't we need N times of $K \geq e^{\frac{1}{N} \sum I(u_{i};\mathcal{T})}$ data to make every $I_{N C E}\left(c_{i} ; \mathcal{T}\right)$ to be a tight bound?**
> > >
> > > ANS:
> > >
> > > To let every $I_{NCE}(c_{i},\mathcal{T})$ to be a tight bound, we just need $ K \geq e^{ \frac{1}{N} \sum_{i=1}^{N} I(u_{i}; \mathcal{T})}$ in total rather than $N e^{\frac{1}{N} \sum_{i=1}^{N} I(u_{i}; \mathcal{T})}$.
> > >
> > > Here, we explain it in detail:
> > >
> > > To make the $I_{NCE}(c_{i};\mathcal{T})$ be a tight bound of $I(c_{i};\mathcal{T})$, we need
> > >
> > > $I_{N C E}\left(c_{i} ; \mathcal{T}\right) \leq I(c_{i} ; \mathcal{T}) \leq \log K$
> > >
> > > Therefore, to make $I_{NCE}(c_{i};\mathcal{T})$ be a tight bound, we need at least
> > >
> > > $\log K \geq I(c_{i};\mathcal{T})$
> > >
> > > Since $I(c_{i};\mathcal{T}) \geq I(u_{i};\mathcal{T})$ (see detailed derivation in Equation 3 in our paper), under the independent assumption, $c_{i}$ is independent to the other confounders $u_{j}(j!=i)$, we need
> > >
> > > $\log K  \geq I(c_{i};\mathcal{T}) \geq I(u_{i};\mathcal{T})$
> > >
> > >
> > >
> > > Thus, to let every $I(c_{i};\mathcal{T})$ be a tight bound, we need
> > > $\sum_{i=1}^{N} I(u_{i} ; \mathcal{T}) \leq \sum_{i=1}^{N} I(c_{i} ; \mathcal{T}) \leq N \log K$
> > >
> > > Therefore, the amount of data K just need to satisfy
> > >
> > > $\log K \geq \frac{1}{N} \sum_{i=1}^{N} I(u_{i}; \mathcal{T})$
> > >
> > > $ K \geq e^{\frac{1}{N} \sum_{i=1}^{N} I(u_{i}; \mathcal{T})}$
> > >
> > >
> > >
> > > [1] David Barber and Felix Agakov. The IM algorithm: A variational approach to information maximization, 2003.
> > >
> > > [2] Oord A, Li Y, Vinyals O. Representation learning with contrastive predictive coding, 2018.
> > >
> > > [3] Chris Cremer, Quaid Morris, and David Duvenaud. Reinterpreting importance-weighted autoencoders, 2017.

---

> ### Author Response · Authors · 2022-08-02
> **Response to Reviewer oNNM (Part 2/3)**
>
> **Q2: 1) Even if the number of confounders increases, the true mutual information  does not$I(c ; \mathcal{T})$. 2)It shows inconsistency to regard $I_{N C E}\left(c ; \mathcal{T}\right)$ and $I_{N C E}\left(c_{i} ; \mathcal{T}\right)$ as having the same upper bound.
> The following inequality also holds in the same setting. $\sum I_{N C E}\left(c_{i} ; \mathcal{T} \mid K\right)=I_{N C E}(c ; \mathcal{T} \mid K) \leq \log K$.**
>
> ANS:
>
> We explain Theorem 1 in detail below:
>
> 1) As the number of confounders increases, although the true mutual information $I(c ; \mathcal{T})$ does not increase, the necessary condition of $I_{NCE}$ to be a tight lower bound of $I_{NCE}$ becomes more difficult to satisfy, and the demand of data  increases significantly.
>
> As for an entangled context, the necessary condition of the InfoNCE lower bound $I_{NCE}(c ; \mathcal{T})$ to be a tight bound is
>
> $$I_{N C E}\left(c ; \mathcal{T}\right) \leq I\left(c ; \mathcal{T}\right) \leq \log K$$
>
> Since $I(c ; \mathcal{T}) \geq \sum_{i=0}^{N} I\left(u_{i} ; \mathcal{T}\right)$, to let the above condition satisfied, the amount of data $K$ must satisfy
>
> $$\log K \geq \sum_{i=0}^{N} I\left(u_{i} ; \mathcal{T}\right)$$
>
> $$K \geq e^{\sum_{i=0}^{N} I(u_{i} ; \mathcal{T})}$$
>
> Therefore, if the number of confounders increases, then the demand for data will grow exponentially.
>
> When data is not rich enough, the nesseray condition may not be satisfied. The InfoNCE lower bound $I_{NCE}(c ; \mathcal{T})$ may be loose, that is $I_{NCE}(c ; \mathcal{T})$ may be much smaller than the true mutual information $I(c ; \mathcal{T})$, thus the MI optimization based on $I_{NCE}(c ; \mathcal{T})$ will be severely affected.
>
> 2) **Clarification:**
> $I_{N C E}\left(c ; \mathcal{T}\right)$ and $I_{N C E}\left(c_{i} ; \mathcal{T}\right)$ have different upper bound.
>
> $I_{N C E}\left(c ; \mathcal{T}\right)$ is the lower bound of $I\left(c ; \mathcal{T}\right)$, and the necessary condition of $I_{N C E}\left(c ; \mathcal{T}\right)$ to be a tight bound of $I\left(c ; \mathcal{T}\right)$ is
>
> $$I_{N C E}\left(c ; \mathcal{T}\right) \leq I\left(c ; \mathcal{T}\right) \leq \log K$$
>
> $I_{N C E}\left(c_{i} ; \mathcal{T}\right)$ is the lower bound of $I\left(c_{i} ; \mathcal{T}\right)$ and the necessary condition of $I_{N C E}\left(c_{i} ; \mathcal{T}\right)$ to be a tight bound of  $I\left(c_{i} ; \mathcal{T}\right)$ is
>
> $$I_{N C E}\left(c_{i} ; \mathcal{T}\right) \leq I\left(c_{i} ; \mathcal{T}\right) \leq \log K$$
>
> As for disentangled context  $c=\{c_{1},c_{2},\cdots,c_{N}\}$, we then derive the necessary condition of $I(c,\mathcal{T})$ to be a tight lower bound of $I(c,\mathcal{T})$:
>
> With the assumption that the contexts $\{c_{1},c_{2},\cdots,c_{N}\}$ are independent to each other, then $I(c ; \mathcal{T})$ could be derived as $\sum I\left(c_{i} ; \mathcal{T}\right)$. Therefore, under the confounder independent assumption, let $I_{NCE}(c ; \mathcal{T})$ be a tight bound is only necessary to let every $I_{NCE}(c_{i} ; \mathcal{T})$ to be a tight bound.
>
> If every $I_{NCE}(c_{i} ; \mathcal{T})(i=1,2,\ldots,N)$ is a tight bound, then we have
>
> $$I_{N C E}\left(c_{i} ; \mathcal{T}\right) \leq I\left(c_{i} ; \mathcal{T}\right) \leq \log K$$
>
> under the confounder independent assumption, we have
>
> $$ \sum I_{N C E}\left(c_{i} ; \mathcal{T}\right) \leq   \sum I\left(c_{i} ; \mathcal{T}\right) \leq N \log K$$
>
> $$I_{N C E} \left(c ; \mathcal{T}\right) = \sum I_{N C E}\left(c_{i} ; \mathcal{T}\right) \leq I\left(c ; \mathcal{T}\right) = \sum I\left(c_{i} ; \mathcal{T}\right) \leq N \log K$$
>
> Thus, the necessary condition of  $I_{N C E} \left(c ; \mathcal{T}\right)$ to be a tight bound of $I \left(c ; \mathcal{T}\right)$ could be relaxed to
>
> $$I_{N C E} \left(c ; \mathcal{T}\right)  \leq I\left(c ; \mathcal{T}\right)  \leq N \log K$$
>
> Therefore, by decomposing the MI estimation under the confounder independent assumption, the demand of the amount $K$ of data could be reduced from  $K \geq e^{I(c ; \mathcal{T})}$ to $K \geq e^{\frac{1}{N} I(c; \mathcal{T})}$.
>
> And with $I(c ; \mathcal{T}) \geq \sum_{i=0}^{N} I\left(u_{i} ; \mathcal{T}\right)$, specificly, the the amount $K$ of data could be reduced from  $K \geq e^{\sum_{i=0}^{N} I\left(u_{i} ; \mathcal{T}\right)}$ to $K \geq e^{\frac{1}{N} \sum_{i=0}^{N} I\left(u_{i} ; \mathcal{T}\right)}$.

---

> ### Author Response · Authors · 2022-08-02
> **Response to Reviewer oNNM (Part 1/3)**
>
> Thanks for your suggestions. We carefully list the derive process in detail and provide more explanations of Theorem 1 to address your concerns as follows:
>
> **Q1:In Appendix A,how does the first equility $q(y \mid x, y_{2: K})=p(y) K w_{y}$ hold? What is $w_{y} $?**
>
> ANS:
>
> Sorry for the confusion. Here, we explain the derivation process thoroughly:
>
> According to the Barber and Agakov's variational lower bound[1], the mutual information  $I(x ; y)$ between $x$ and $y$ can be bounded as follows:
>
> \begin{equation}
> I(x ; y)=E_{p(x, y)} \log \frac{p(y \mid x)}{p(y)} \geq E_{p(x, y)} \log \frac{q(y \mid x)}{p(y)}
> \end{equation}
> where $q$ is an arbitrary distribution.
>
> Specifically, $q(y \mid x)$ is defined by independently sampling a set of examples $[y_{1}, \ldots, y_{K}]$ from a proposal distribution $\pi(y)$ and then choosing $y$ from $[y_{1}, \ldots, y_{K}]$ in proportion to the importance weights
>
>
> $$w_{y}=\frac{e^{\psi(x, y)}}{\sum_{k} e^{\psi\left(x, y_{k}\right)}}$$
>
> ,where $\psi$ is a function that takes $x$ and $y$ and outputs a scalar. According to [2], by setting the proposal distribution as the marginal distribution $\pi(y) \equiv p(y)$, the unnormalized density of $y$ given a specific set of samples $y_\{2: K}=[y_{2}, \ldots, y_{K}]$ and $x$ is:
>
> \begin{equation}
> q\left(y \mid x, y_{2: K}\right)=p(y) \cdot \frac{K \cdot e^{\psi(x, y)}}{e^{\psi(x, y)}+\sum_{k=2}^{K} e^{\psi\left(x, y_{k}\right)}}=Kp(y)w_{y}
> \end{equation}
> where $K$ denotes the numbers of samples.
>
> According to [3], the expectation of $q\left(y \mid x, y_{2: K}\right)$ with respect to resampling of the alternatives $y_{2: K}$ from $p(y)$ produces a normalized density:
> \begin{equation}
> \bar{q}(y \mid x)=E_{p\left(y_{2: K}\right)}\left[q\left(y \mid x, y_{2: K}\right)\right]
> \end{equation}
>
> The $I_{N C E}(x ; y \mid E, K)$ is a typo, it should be $I_{\mathrm{NCE}}(x ; y \mid \psi, K)$.
>
> We have refined the proof process based on your kind suggestions and added relevant details, see Appendix A in the revised version.
>
> [1] David Barber and Felix Agakov. The IM algorithm: A variational approach to information maximization, 2003.
>
> [2] Oord A, Li Y, Vinyals O. Representation learning with contrastive predictive coding, 2018.
>
> [3] Chris Cremer, Quaid Morris, and David Duvenaud. Reinterpreting importance-weighted autoencoders,
> 2017.

---

> ### Comment · Reviewer_oNNM · 2022-08-09
> **Remaining concerns**
>
> Thank you for your detailed response to my questions.
> However, I still have remaining concerns on Theorem 1.
>
> To make each $I_{NCE}(c_i;\mathcal T)$ be a tight bound, we need $K \ge e^{I(u_i; \mathcal T)}$ amount of data.
> We are going to optimize $I_{NCE}(c_i;\mathcal T)$ by using separate negative and positive pairs for each context $i$.
> Then, don't we need $N$ times of $K \ge e^{I(u_i; \mathcal T)}$ amount of data?
>
> The authors clarified that $I_{NCE}(c;\mathcal T)$ and $I_{NCE}(c_i;\mathcal T)$ have different upper bound. And the authors are using the same upper bound $log K$ for both cases. I understood this $K$ as an arbitrary variable that represents the amount of data we need. So, the same $K$ does not mean the same amount of data, right? Then summing up $K$ in the proof of Theorem 1 does not make sense.

---

### Author Response · Authors · 2022-08-02
**Overall Response**

We sincerely appreciate all reviewers' time and efforts in reviewing our paper. We are glad to find that reviewers generally recognized our key contributions and clear presentation of our paper:

**Contributions**.
- **Method**: This paper provides a decomposed mutual information optimization framework for improving context-aware meta RL in the environment with multiple confounders that impact the transition dynamics[RkbU, rYrP]. It is the first method to learn disentangled context by directly exploiting the confounders independent assumption [rYrP]. The idea is novel, intuitive and interesting[oNNM,1VJF].
- **Experiment**: Extensive experiments show the effectiveness of the proposed method [RkbU]. The proposed method achieve better performance than the baselines[rYrP,1VJF] and performs well against an ablation that does not learn disentangled context vectors [rYrP]. Experiments are comprehensive, and the results are impressive[oNNM].
- **Presentation**. The paper is well written and clear to understand [oNNM,RkbU]. The method itself is clearly described and is easy to follow [rYrP]. The figures in this paper are very clear and very well [RkbU].

Also, we thank all reviewers for their valuable and constructive suggestions, which help us a lot in improving our paper. In addition to the pointwise responses below, we have updated our paper in the revised version to incorporate the insightful suggestions of the reviewers:

**Experiments.**
- Following Reviewer RkbU's suggestions,
we add a **comparison experiment to compare DOMINO and RIA**[1]. (see Figure 8, Appendix E.2, page21).

- Following Reviewer RkbU's and reviewer rYrP's suggestions,
we construct the **sensitivity analysis experiment of hyper-parameter N** to how dependent the performance is on setting N (see Figure 9, Appendix E.3, page 21).

- Following Reviewer 1VJF 's suggestions,
we provide **visualization analysis of each latent context** to show that the learned context vectors are disentangled well and each latent context captures one of the confounders of the environment(see Appendix E.3, page21, revised version paper).

- Following Reviewer rYrP's suggestions,
we add a **Welch t-test** between the proposed method and the baselines.

**Derivation.**

we carefully provide more detailed proof of Lemma 1  and the explanation of Theorem 1 (see Appendix A.1 and A.2, page 14-15, revised version paper).

[1] Guo J, Gong M, Tao D. A Relational Intervention Approach for Unsupervised Dynamics Generalization in Model-Based Reinforcement Learning[C]//International Conference on Learning Representations. 2022.

We hope our pointwise responses below could clarify all reviewers confusion and alleviate all concerns. We thank all reviewers’ time again and we always ready to solve your concerns.

---

### Author Response · Authors · 2022-08-06
**Submit the third revision of our paper**

We again appreciate the reviewers for their discussion and valuable comments.

We submitted the second revision of our paper.
- The first revision: **red**
- The second revision: **blue**
- The third revision: **purple**

We are looking forward to discussing any concerns of our paper.

We are ready to address your further concerns and revise the paper if the reviewers suggest anything that can improve the quality of our paper.

---

### Author Response · Authors · 2022-08-09
**Summary of our rebuttal and discussion**

We sincerely appreciate all reviewers' and ACs' time and efforts in reviewing our paper. We truly thank you all for the insightful and constructive suggestions, which helped improve the quality of our paper. We appreciate all reviewers have actively discussed with us in-depth and read our responses carefully, which gave us a great submission experience.
We genuinely appreciate the positive 6-7-6-6 evaluation from reviewers oNNM, rYrP, 1VJF, and RkbU.

Here is a summary of our updates:

[Additional Experiments] As suggested by reviewers RkbU and rYrP's, we conducted comparison experiments to compare the performance of DOMINO and RIA and the sensitivity analysis experiment of hyper-parameter N. As suggested by reviewer 1VJF, we provide a visualization analysis of each latent context to show that the learned context vectors are disentangled well. All additional results consistently validate the effectiveness of our proposal.

[Detailed Derivation] As suggested by reviewer oNNM, we provide more detailed proof of Lemma 1 and the explanation of Theorem 1.

[Writing] We owe many thanks to all the reviewers' helpful writing suggestions. We provide clearer definitions, more accurate expressions, and more discussions on the related works in our revision.

We really thank all reviewers' and ACs' time and efforts again.

Best wishes,

Authors

---

### Meta-Review · Area_Chair_SkT8 · 2022-08-24

**Recommendation:** Accept
**Confidence:** Certain

**Metareview:**

This paper proposes DOMINO, an optimization framework, for contextual meta reinforcement learning. The reviewers generally agree that the paper is well written, the idea is novel and interesting, the evaluation is comprehensive and the results are impressive. Reviewers also raised a few concerns in the initial reviews, such as the proof of Lemma 1 and Theorem 1, and the mathematical definitions. Throughout the discussion phase, most of these concerns were sufficiently addressed, and the review scores were increased accordingly. Overall, the quality of the revised paper has improved significantly during the rebuttal. Thus, I recommend accepting this paper. Please incorporate the remaining reviewers' suggestions in the future version of this paper.

**Award:**

No

---

### Decision · Program_Chairs · 2022-09-14

Accept